# A quantitative framework for whole-body coordination reveals specific deficits in freely walking ataxic mice

Ana S Machado[†], Dana M Darmohray[†], João Fayad, Hugo G Marques, Megan R Carey*

Champalimaud Neuroscience Programme, Champalimaud Centre for the Unknown, Champalimaud Foundation, Lisbon, Portugal

**Abstract** The coordination of movement across the body is a fundamental, yet poorly understood aspect of motor control. Mutant mice with cerebellar circuit defects exhibit characteristic impairments in locomotor coordination; however, the fundamental features of this gait ataxia have not been effectively isolated. Here we describe a novel system (LocoMouse) for analyzing limb, head, and tail kinematics of freely walking mice. Analysis of visibly ataxic *Purkinje cell degeneration* (*pcd*) mice reveals that while differences in the forward motion of individual paws are fully accounted for by changes in walking speed and body size, more complex 3D trajectories and, especially, inter-limb and whole-body coordination are specifically impaired. Moreover, the coordination deficits in *pcd* are consistent with a failure to predict and compensate for the consequences of movement across the body. These results isolate specific impairments in whole-body coordination in mice and provide a quantitative framework for understanding cerebellar contributions to coordinated locomotion.

*For correspondence: megan. carey@neuro.fchampalimaud.org

[†]These authors contributed equally to this work

Competing interests: The authors declare that no competing interests exist.

## Introduction

Our ability to engage in even the simplest motor tasks requires us to coordinate our movements in space and time across the body. For example, during walking, leg, arm, trunk, and head movements need to be coordinated to achieve a stable, smooth, and efficient gait. The cerebellum is critical for coordinated locomotion; cerebellar damage leads to characteristic gait ataxia across species. Exactly how the cerebellum contributes to motor coordination, however, remains controversial.

Mice present several advantages for understanding the role of the cerebellum in locomotor coordination. In addition to their amenability to genetic circuit dissection, their small size makes it possible to analyze even unrestrained, relatively complex whole-body actions within a laboratory setting. Moreover, several spontaneous mutants have been identified based on visible gait ataxia (*Mullen et al., 1976*; *Walter et al., 2006*; *Lalonde and Strazielle, 2007*; *Cendelin, 2014*). These mutants exhibit abnormal cell patterning within the cerebellum (*Lalonde and Strazielle, 2007*; *Brooks and Dunnett, 2009*; *Kim et al., 2009*; *Sheets et al., 2013*; *Cendelin, 2014*). While traditional gait analyses in ataxic mutants have reported a variety of impairments in individual limb movements and interlimb coordination (*Fortier et al., 1987*; *Wang et al., 2006*; *Vinueza Veloz et al., 2014*), many of these findings are not specific to ataxia and could represent secondary consequences of changes in walking speed (*Batka et al., 2014*). Therefore it has not been clear to what extent exisiting analyses of mouse locomotion capture the essence of the coordination deficits of ataxic mice that are so visible to the human eye (*Cendelin et al., 2010*).

Here we describe LocoMouse, a novel system for automated, markerless, 3D tracking of locomotor kinematics in freely walking mice. We used LocoMouse to establish a quantitative framework for locomotor coordination in mice and analyze the deficits of visibly ataxic *Purkinje cell degeneration*

**eLife digest** Though it seems simple, walking is a complex activity. The arms, legs, body, and head all need to work together. A part of the brain called the cerebellum helps to coordinate the movements of different body parts allowing both simple and complex tasks to be carried out smoothly. But it is not known exactly how the cerebellum coordinates body movements.

Studies of mice have helped shed some light on the coordination of movement. Several mutations that naturally occur in mice can cause them to walk abnormally. These mutations often cause changes in the cerebellum. Neuroscientists studying these mutant mice often use balance beams or other challenging tasks to compare their coordination with typical mice. But studies attempting to measure specific changes in walking movements under natural conditions have yielded conflicting results.

Now, Machado, Darmohray et al. have demonstrated that an automated movement-tracking system can capture specific aspects of coordination in freely walking mice. The system, called 'LocoMouse', uses high-speed cameras and computers to document and analyze the paw, nose, and tail movements of mice walking through a glass hallway. In the experiments, the system was used to compare typical mice with mice that have a mutation affecting the cerebellum that causes them to walk abnormally.

Unexpectedly, Machado, Darmohray et al. found that forward steps of the mutant mice are comparable to the steps of the typical mice, if you account for the fact that the mutant mice are smaller and slower. Instead, however, the mutants were found to have specific difficulties coordinating movement across the body. The movements of the mutant mice's front and hind paws, for example, did not follow the same coordinated pattern as the typical mice. The mutant mice also swung their head and tail in an exaggerated way. Machado, Darmohray et al.'s analysis revealed that these movements likely resulted from a failure of the cerebellum of the mutant mice to predict and compensate for the motion of the rest of the body. Other scientists will now likely use the LocoMouse system to study mouse movements and how the brain controls them.

*(pcd)* mutants. Surprisingly, we find that the forward motion of individual paws is normal in *pcd*—apparent differences in basic stride parameters and forward trajectories disappeared once changes in body size and walking speed were taken into account. In contrast, 3D paw trajectories, interlimb and whole-body coordination were specifically impaired. Moreover, the prominent nose and tail oscillations observed in *pcd* were successfully modeled as passive consequences of the forward motion of the hind limbs, suggesting the absence of a mechanism that normally predicts and compensates for movements of other parts of the body. Taken together, these results suggest a specific failure to predict the consequences of movement across joints, limbs, and body, and are consistent with the hypothesis that the cerebellum provides a forward model for motor control (*Bastian et al., 1996*; *Ebner and Pasalar, 2008*; *Kennedy et al., 2014*).

## Results

### LocoMouse: a system for quantifying locomotor coordination

The noninvasive, markerless LocoMouse system (*Figure 1*) uses high-speed cameras and machine learning algorithms to automatically detect and track the position of paws, nose, and tail in 3D with high (2.5 ms) temporal resolution.

Mice walked across a glass corridor, 66.5 cm long and 4.5 cm wide (*Figure 1A*). A mirror was placed at 45 deg under the mouse, so that a single high-speed camera (AVT Bonito, 1440x250 pixels @400 frames per second) recorded both bottom and side views. Individual trials consisted of single crossings of the corridor. Mice freely initiated trials by walking back and forth between two dark 'home' boxes on each end of the corridor. Data collection was performed in LABVIEW and was automatically triggered by infrared sensors that detected when the mouse entered and exited the corridor.

After processing the images to subtract the background and correct for mirror and lens distortions, we applied a machine learning algorithm (*Figure 1B*) to identify and track all four paws, snout, and 15 tail segments in both bottom and side views for each trial (*Figure 1C*; *Video 1*; see 'Materials and

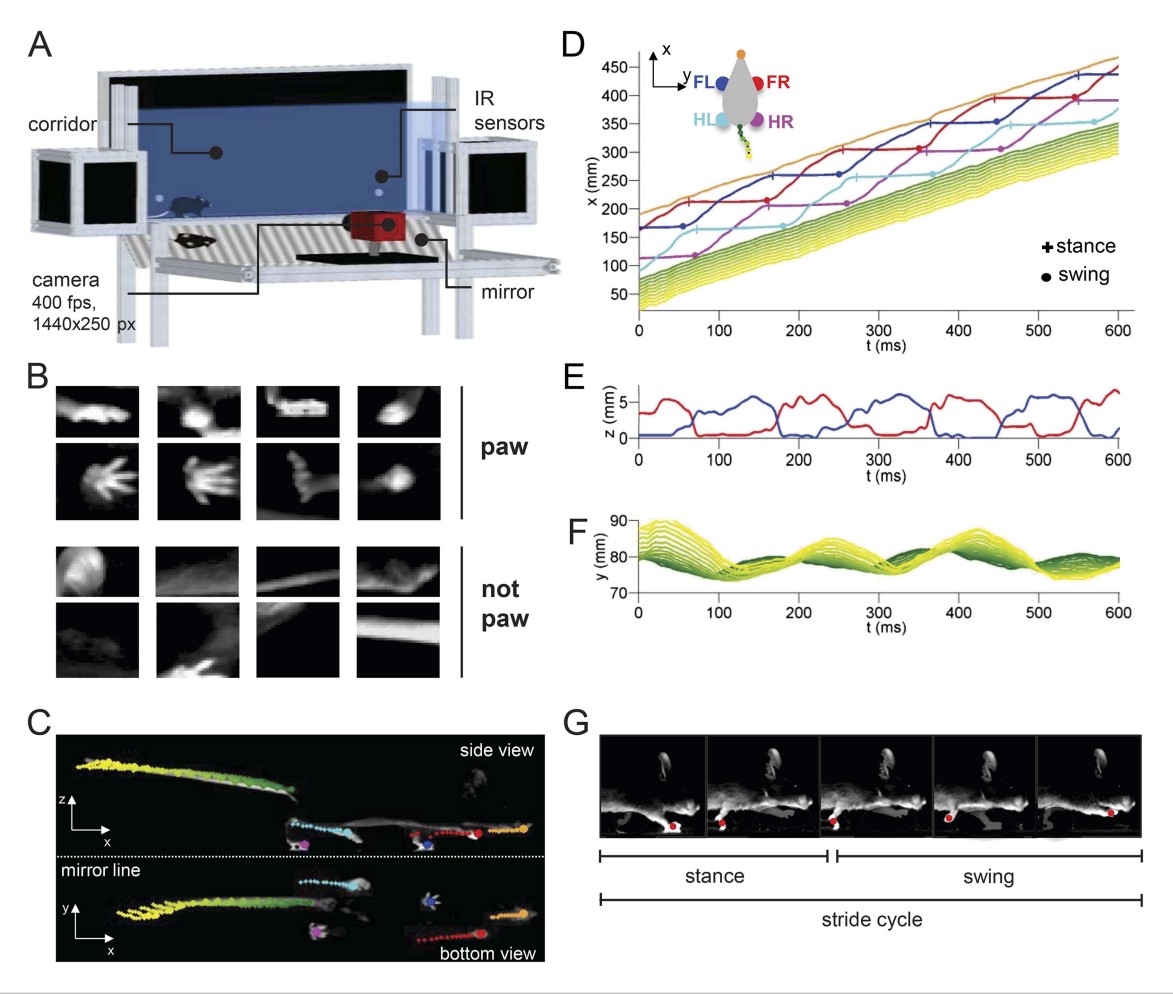

**Figure 1**. LocoMouse system for analyzing mouse locomotor coordination. (**A**) LocoMouse apparatus. The mouse walks freely across a glass corridor with mirror below at a 45° angle. A single high-speed camera captures side and bottom views at 400 fps. Infrared (IR) sensors trigger data collection. (**B**) Machine learning algorithms identify paws, nose and tail segments and track their movements in 3D. Example 'paw' and 'not paw' training images for SVM (Support Vector Machine) feature detectors are shown for side and bottom views. (**C**) Continuous tracks are obtained by post-processing the feature detections with a Multi-Target Tracking algorithm. (**D**) Continuous forward trajectories (x position vs time) for paws, nose, and tail. The inset illustrates the color code used throughout the paper to identify individual features. (**E**) Continuous vertical (z) trajectories of the two front paws. (**F**) Side-to-side (y) position of proximal (green) to distal (yellow) tail segments vs time. (**G**) Individual strides were divided into swing and stance phases for further analysis. Further validation of tracking algorithm is presented in *Figure 1—figure supplement 1*.

The following figure supplement is available for figure 1:

**Figure supplement 1**. LocoMouse tracking validation.

methods'). We then extracted the continuous forward (x), side-to-side (y), and vertical (z) trajectories for each feature from each movie (*Figure 1D–F*). The stride cycles of all four paws were automatically divided into swing and stance phases for subsequent analysis (*Figure 1G*). Validation of the tracking is provided in *Figure 1—figure supplement 1*.

## Paw stride parameters vary consistently with walking speed and body size

We first analyzed basic stride parameters for individual limbs of wildtype control mice (*Figure 2*). Parameters such as stride length (mm), cadence (strides/s), swing velocity (m/s), and stance duration (ms), along with the mouse's walking speed, were measured for each stride. The data were highly

**Video 1.** Automated, high-resolution locomotion tracking in freely walking mice. High-speed (400 fps) video of a mouse crossing the LocoMouse corridor, displayed at 30 fps. Side and bottom (via mirror reflection) views of the mouse are captured in a single camera. Top: Raw video of a wild-type mouse. Bottom: Same video with the output of the machine learning tracking overlaid: nose (orange circle), paws (red: front right; blue: front left; magenta: hind right; cyan: hind left) and tail segments (green-to-yellow gradient circles).

variable (*Figure 2A*; see also *Figure 2—figure supplement 1*). The walking speed of the mice was similarly variable (*Figure 2B*). We therefore sorted all strides for individual mice into speed bins in a stridewise manner and analyzed them with respect to the mouse's walking speed.

The median values of stride parameters for each mouse across speed bins are shown in *Figure 2C–F* (dots connected by dashed lines). Each parameter measured, including stride length, cadence, swing velocity, and stance duration (*Figure 2C–F*), varied consistently with the walking speed of the mouse. Cadence and stride length increased with walking speed, indicating that faster walking in mice is associated with longer, more frequent strides (*Clarke and Still, 1999*; *Lalonde and Strazielle, 2007*; *Batka et al., 2014*). These changes, in turn, resulted from linear increases in swing velocity and steep decreases in stance duration with increasing walking speed. Further subdividing the data by the body weight of each animal (*Figure 2—figure supplement 1*) revealed that much of the remaining variability in each parameter could be accounted for by the mouse's body size (*Figure 2*, color-coded by weight).

To quantify the influence of walking speed, body size, and other potential factors on these basic stride parameters, 36,369 strides from an average of 1069 ± 266 strides in 34 mice were analyzed (*Figure 2*). For each parameter, we first linearized the data by fitting appropriate functions (e.g., linear, power) to the data with respect to walking speed (*Figure 2—figure supplement 1*). Then we generated a multilevel linear mixed-effects model that included potential predictor variables speed, weight, body length, age, and gender either alone or in combination and asked to what extent they accurately predicted the measured parameter. This analysis revealed that the value of each stride parameter was readily predicted based solely on walking speed and body weight (*Figure 2—figure supplement 1*). While basic stride parameters also varied with gender and age, these effects were related to differences in body size (*Figure 2—figure supplement 1*); adding neither age nor gender improved the predictions once body size was taken into account. The resulting best-fit models are plotted as thick lines in *Figure 2C–F*. These results indicate that the equations in *Figure 2—figure supplement 1* provide quantitative predictions of paw stride parameters for mice of a given size, walking at a particular speed.

## Purkinje cell degeneration (*pcd*) mice

The systematic analysis of stride parameters across mice presented above provided a starting point for quantifying locomotor deficits of ataxic mice. The Purkinje cell degeneration (*pcd*) mouse is a recessive mutant characterized by complete post-natal degeneration of cerebellar Purkinje cells and subsequent partial loss of cerebellar granule cells ([*Chen et al., 1996*; *Le Marec and Lalonde, 1997*; *Lalonde and Strazielle, 2007*; *Cendelin, 2014*]; The gene affected encodes ATP/GTP binding protein 1 [*Fernandez-Gonzalez et al., 2002*]). *Pcd* mice can be easily identified by eye based on their ataxic, uncoordinated movements (*Mullen et al., 1976*; *Le Marec and Lalonde, 1997*). *Pcd* mice exhibit impaired rotarod performance and deficits in eyelid conditioning that have been attributed to their cerebellar abnormalities (*Chen et al., 1996*; *Le Marec and Lalonde, 1997*). Perhaps surprisingly, given the severity of their anatomical phenotype, the motor deficits of *pcd* mice are relatively mild compared to other spontaneous ataxic mutants (*Lalonde and Strazielle, 2007*; *Le Marec and Lalonde, 1997*).

## Changes in stride parameters are predicted by changes in walking speed and body size

*Pcd* mice were visibly ataxic when walking on the LocoMouse setup (*Video 2*). Consistent with previous studies of cerebellar ataxia in mice (*Fortier et al., 1987*; *Wang et al., 2006*; *Cendelin et al., 2010*; *Vinueza Veloz et al., 2014*), comparing the basic stride parameters of visibly ataxic *pcd* mice

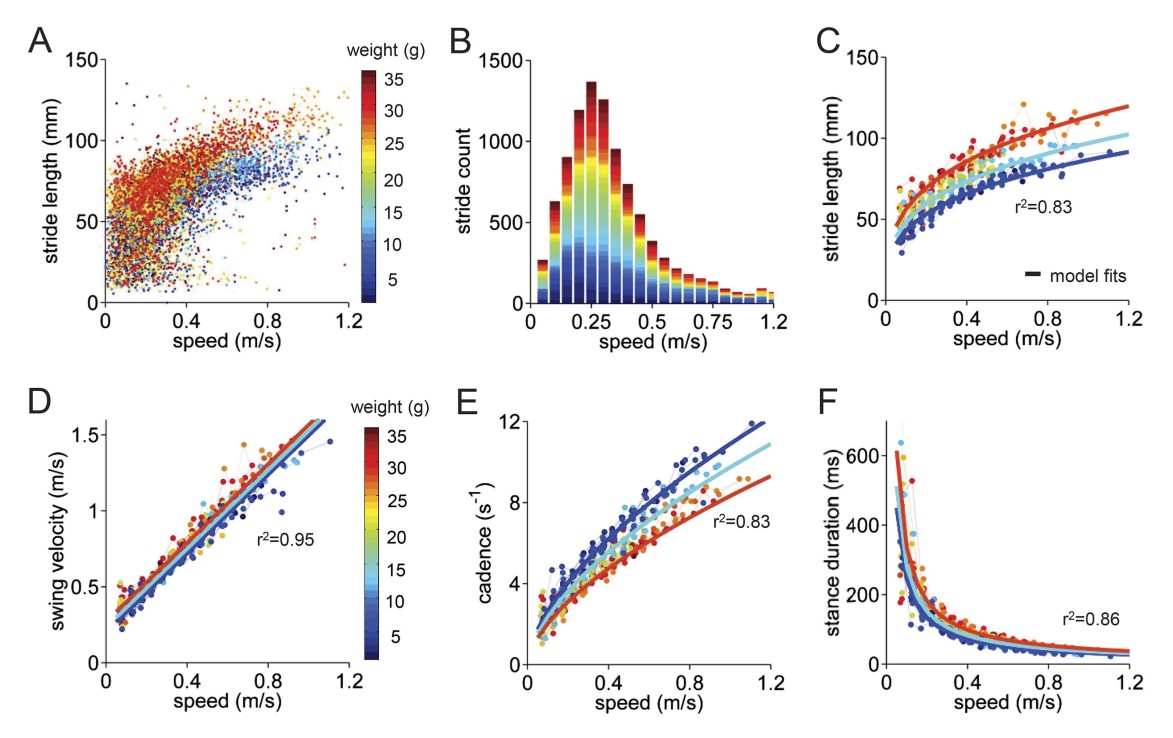

**Figure 2**. Basic stride parameters can be predicted using only walking speed and body size. (**A**) Stride length vs walking speed for 9602 individual strides of the front right paws of 34 wildtype mice are color-coded by weight for each individual animal. More information on the mice can be found in *Figure 2—figure supplement 1* and *Figure 2—source data 1*. (**B**) Histogram of average walking speeds for each stride from (**A**). Strides are divided into speed bins of 0.05 m/s. (**C–F**) Stride length, cadence (1/stride duration), swing velocity, and stance duration vs walking speed, respectively. For each parameter, speed-binned median values are shown for each animal (solid circles, color coded by weight). Data for each animal are connected across speeds with a thin dotted line. Thick lines are the output of the linear mixed-effects model, for 3 example weights across walking speeds (blue: 9g, cyan: 19g, red: 33g). Marginal R-squared values for each linear mixed model are shown for each parameter. All raw data for *Figure 2* and *Figure 2—figure supplement 1* can be found in *Figure 2—source data 1*.

The following source data and figure supplement are available for figure 2:

**Source data 1**. Individual limb gait parameters, walking speed, and mouse metrics used to generate the linear mixed effects model in *Figure 2*.

**Figure supplement 1**. Using linear mixed effects models to predict basic stride parameters.

with littermate control mice revealed that the strides of *pcd* mice were, overall, quite different (*Figure 3A–D*). Stride lengths were shorter (*Figure 3B*, purple shadows), even when changes in walking speed (*Figure 3A*) were taken into account. Cadence and stance durations were also altered (*Figure 3C,D*, purple shadows).

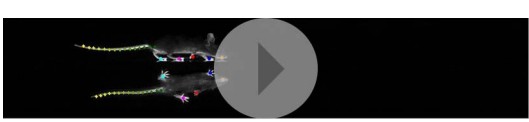

**Video 2.** Purkinje cell degeneration mice are visibly ataxic. *Purkinje cell degeneration (pcd)* mouse crossing the LocoMouse corridor. *Pcd* mice are smaller and walk more slowly than controls. They lift their paws higher and have altered patterns of interlimb coordination. The nose and tail oscillate laterally and vertically.

Since *pcd* mice, like many ataxic animals, are smaller than controls (*Figure 3—figure supplement 1*), and given that they walk more slowly (*Figure 3A*), we asked to what extent the altered stride parameters in *pcd* could be accounted for simply by changes in body size and walking speed. To do this we used the equations derived from the linear mixed-effects models in *Figure 2* to predict stride parameters across walking speeds for mice the size of the *pcd* mice and their littermates. The models accurately predicted stride parameters for the

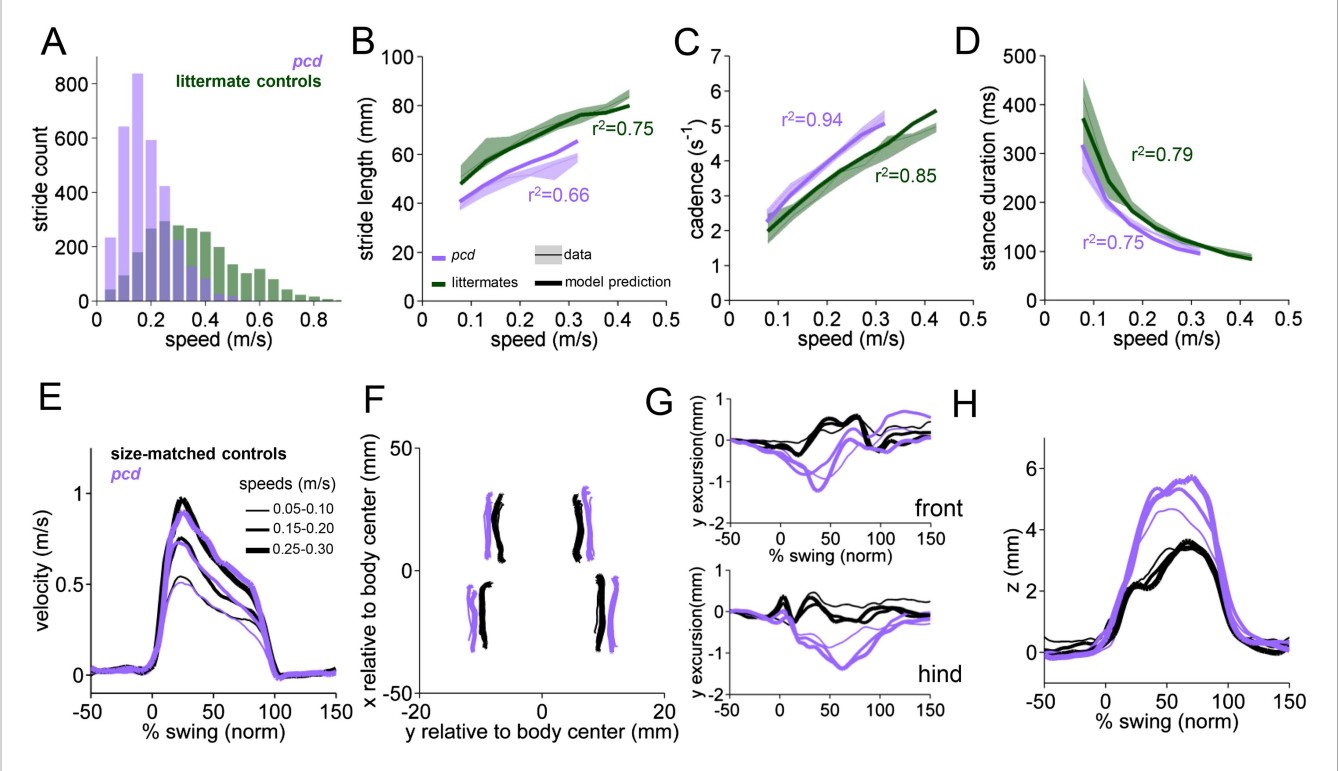

**Figure 3**. Differences in forward paw trajectories in *pcd* can be accounted for by walking speed and body size; impairments are restricted to off-axis movement. (**A**) Histogram of walking speeds, divided into 0.05 m/s speed bins for *pcd* (purple N = 3 mice; n = 3052 strides) and littermate controls (green, N = 7; n = 2256 strides). (**B–D**) Stride length (**B**), cadence (**C**, 1/stride duration) and stance duration (**D**) vs walking speed for *pcd* mice (purple) and littermate controls (green). For each parameter, the thin lines with shadows represent median values ±25th, 75th percentiles. Thick lines represent the predictions calculated using the mixed-effect models described in *Figure 2* and *Figure 2—figure supplement 1* (including speed and weight as predictor variables). (**E**) Average instantaneous forward (x) velocity of FR paw during swing phase for *pcd* (purple) and size-matched controls (black). Line thickness represents increasing speed. (**F**) x-y position of four paws relative to the body center during swing. (**G**) y-excursion for front and hind paws, relative to body midline. (**H**) Average vertical (z) position of FR paw relative to ground during swing.

The following figure supplements are available for figure 3:

**Figure supplement 1**. Basic stride parameters for *pcd* are are not different from their size-matched controls.

**Figure supplement 2**. 3D paw trajectories for wildtype controls and *pcd*.

littermates, which were not visibly ataxic (*Figure 3B–D*, green: thick lines represent model predictions). Surprisingly, we also found that the models accurately predicted stride parameters of *pcd* mice (*Figure 3B–D*, purple). Thus, although stride parameters of *pcd* mice were different overall from controls (*Figure 3B–D*, purple vs green shadows), they were comparable to those predicted for control mice of similar body size walking at similar speeds (*Figure 3B–D*, the thick lines representing the model predictions fall on top of the data in the shadows). Moreover, a direct comparison between stride parameters for *pcd* mice and size-matched controls walking at the same speeds revealed no difference between the two groups (*Figure 3—figure supplement 1*) (stride length $F_{(14,1)} = 0.70$, p = 0.42; cadence $F_{(14,1)} = 0.004$, p = 0.95; stance duration $F_{(14,1)} = 1.89$, p = 0.17).

Next we investigated the possibility that there could be changes in variability of stride parameters in pcd that were not apparent in the averaged data. Analysis of the coefficient of variation revealed that swing length variability was unchanged in pcd compared to size matched controls ($F_{(81,1)} = 0.14$, p = 0.0071). Surprisingly, both cadence and stance duration were less variable in *pcd* (cadence: $F_{(80,1)} = 6.90$, p < 0.05; stance duration: $F_{(80,1)} = 6.90$, p < 0.05).

Taken together, these results demonstrate that basic stride parameters, although altered in *pcd*, do not capture the ataxic symptoms of *pcd* mice, and highlight the importance of accounting for

walking speed (*Koopmans et al., 2007*; *Cendelin et al., 2010*; *Batka et al., 2014*) and using size-matched control animals when analyzing locomotor parameters. For this reason *pcd* animals are compared with size-matched controls from here on (*Figure 3—figure supplement 1*).

## Strides of *pcd* mice show specific impairments in off-axis paw trajectories

It has been previously hypothesized that detailed analysis of paw trajectories would capture the gait abnormalities of ataxic mice like *pcd*, but detailed 3D paw kinematics have not been described for mice. We analyzed the continuous 3D paw trajectories for both wildtype and *pcd* mice (*Figure 3E–H*; *Figure 3—figure supplement 2*). Surprisingly, we found that the instantaneous forward paw velocity profiles of *pcd* mice were not distinguishable from those of size-matched controls, across speeds (*Figure 3E*). Paw velocity peaked early during swing and decelerated before stance onset across walking speeds in both control and *pcd* mice (*Figure 3E*; *Figure 3—figure supplement 2*). Peak swing velocities increased with faster walking speeds but did not vary by genotype ($F_{(10.97,1)=}.092$, p = 0.77). There was also no difference in variability of peak swing velocity between genotypes ($F_{(81,1)=}0.27$, p = 0.60). This surprising result reveals that even detailed forward paw trajectories are normal in *pcd* mice, once changes in walking speed and body size are taken into account.

We next examined the horizontal (y) and vertical (z) movements of the paws (*Figure 3F–H*; *Figure 3—figure supplement 2*). Consistent with previous findings of ataxic mice and humans, *pcd* mice exhibited a wider base of support than size-matched control mice (*Figure 3F*) ($F_{(15.51,1)=}42.87$, p <0 0.001). There were also subtle changes in side-to-side (y) paw trajectories ($F_{(13.88,1)=}20.64$, p <0 0.001), especially for front limbs ($F_{(152.95,1)=}12.125$, p <0 0.001; *Figure 3G*). Further, analysis of the vertical (z) trajectories revealed significantly larger vertical displacement of both front and hind paws of *pcd* mice, across speeds ($F_{(66.84,1)=}17.16$, p <0 0.001) (*Figure 3H*; *Figure 3—figure supplement 2*). The variability of this vertical displacement was not different in pcd ($F_{(81,1)=}2.47$, p = 0.12).

Thus, despite the visibly ataxic walking pattern of *pcd* mice, the results of the mixed-effects linear models and the trajectory analyses indicate that the forward motion of the paws was remarkably preserved in *pcd*. Alterations in individual limb movements were restricted to off-axis (horizontal and vertical) trajectories.

## Front-hind interlimb coordination is specifically impaired in *pcd*

Analyses of mouse locomotion that have focused on quantifying the kinds of basic stride parameters presented in *Figure 2* have previously failed to quantitatively capture gait ataxia in visibly ataxic mice (*Cendelin et al., 2010*). We reasoned that this could be because human observers are more sensitive to the patterns of movement across different parts of the body (*Basso et al., 2006*). Therefore we analyzed patterns of interlimb and whole body coordination in both control and *pcd* mice.

In our experiments, wildtype mice walked in a symmetrical trot pattern across speeds—each diagonal pair of limbs moved together and alternated with the other pair (*Figure 4A*, left). According to the terminology of *Hildebrand (1989)*, at slower speeds there was a tendency toward a 'walking trot' (front paws in a diagonal pair touch down just before hind paws and the paws are on the ground more than 50% of the time), while at faster speeds a 'running trot' was observed (diagonal paw pairs strike the ground near-simultaneously and paws are on the ground less than 50% of the time) (*Figure 4—figure supplement 1*). There was no abrupt shift between these gait patterns—stance phases varied smoothly with walking speed and duty cycle (*Figure 4A*, *Figure 4—figure supplement 1*) (cf. *Bellardita and Kiehn, 2015*). We did not observe galloping or bounding even at the highest speeds (cf. *Bellardita and Kiehn, 2015*), probably because the mice freely initiated trials in our experiments, rather than being placed in the corridor by the experimenter at the start of each trial (*Figure 4—figure supplement 1*). For ease of quantification, and because of a lack of categorical gait boundaries in our data, we analyzed interlimb coordination in terms of phase values and support patterns rather than gait patterns.

The normal pattern of interlimb coordination was markedly disrupted in *pcd*, due to specific and consistent changes in the phase relationship between front and hind limbs ($F_{(77.07,1)} = 4.11$, p <0 0.05; *Figure 4A*, right; *Figure 4B*). Importantly, in marked contrast to the front-hind limb coupling, left-right alternation was maintained in *pcd* ($F_{(159,1)} = 0.018$, p=0.89; *Figure 4A*, right: red vs blue and cyan vs magenta; *Figure 4B*, left). Thus, as a result of the de-synchronization of front and hind paw

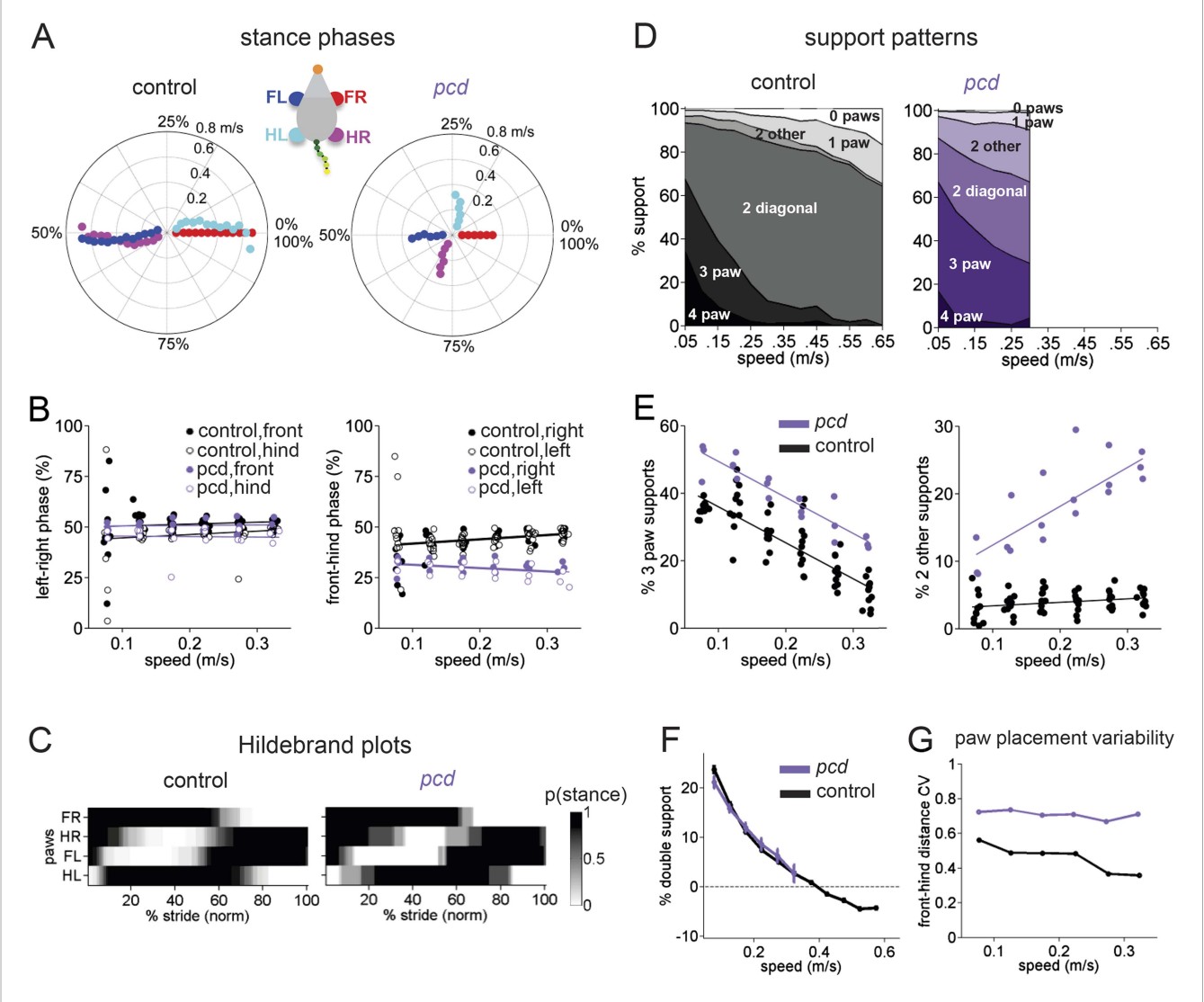

**Figure 4.** Front-hind limb coordination is specifically impaired in *pcd*. (**A**) Polar plots indicating the phase of the step cycle in which each limb enters stance, aligned to stance onset of FR paw (red). Distance from the origin represents walking speed. Left, size-matched control mice (N = 11). Right, *pcd* (N = 3). (**B**) Left-right phase (left) and front-hind (right) phase for individual animals of *pcd* and size-matched controls. Circles show average values for each animal. Lines show fit of linear-mixed effects model for each variable. (**C**) Average Hildebrand plots aligned to FR stance onset for speeds between 0.15 and 0.20 m/s. Grayscale represents probably of stance. (**D**) Area plot of average paw support types as % of stride cycle, across speeds for size-matched controls (left) and pcd (right). (**E**) 3 paw (left) and 2-paw other (right) supports for each animal (circles). Lines show fit of linear-mixed effects model. (**F**) Average ±sem percent double support for hind paws of pcd and size-matched controls. (**G**) Coefficient of variation for paw placement distance (front-hind) for pcd and size-matched controls.

The following figure supplement is available for figure 4:

**Figure supplement 1**. Comparison of gait patterns between control and *pcd* mice.

movements, the diagonal limbs no longer moved in phase with each other, as illustrated in the Hildebrand plots in *Figure 4C*.

Support patterns, or the configuration of paws on the ground at any given time, vary systematically with walking speed (*Górska et al., 1999*). Typically, wildtype mice had two diagonal paws on the ground at any given time (2-paw diagonal support, *Figure 4D*, left), but this ranged from 3 paws on the ground during slow walking to 0 paw supports, or brief periods of flight, during running at higher speeds, due to changes in stance to swing phasing (*Figure 4—figure supplement 1*). *Pcd* mice spent

more time with more paws on the ground (*Figure 4D*, right; *Figure 4E*, left) (3-paw support $F_{(82,1)}$ = 83.57, p < 0.001). Moreover, while % double paw support was the same for *pcd* and size-matched control mice walking at comparable speeds ($F_{(167,1)}$ = 1.06, p = 0.31), the upper limit of *pcd* walking speeds coincided with the transition from positive to negative % double hind limb supports (*Figure 4F*, see 'Materials and methods'). In other words, it appears that the walking speed of *pcd* mice is limited by the need to have at least one hind paw on the ground, for postural stability (*Stolze et al., 2002*).

Despite their slower walking speeds and increased percent of time spent with more paws on the ground, *pcd* mice also showed an increase in unstable support configurations such as non-diagonal 2-paw support (*Figure 4E*, right; *Figure 4—figure supplement 1*) ($F_{(46.78,1)}$ = 7.76, p = 0.01), particularly at higher walking speeds ($F_{(67.62,1)}$ = 115.82, p < 0.001). This increased instability indicates that *pcd* mice are not simply switching to a more stable gait pattern as a compensatory mechanism, but rather, are unable to properly time their front-hind limb movements to generate a stable, efficient gait. Further, the changes in interlimb phasing were consistent—front-hind phasing was not more variable in pcd ($F_{(164,1)}$ = 2.88, p = 0.091), and in fact left-right phasing was even less variable ($F_{(166,1)}$ = 9.70, p = 0.0021).

Spatial patterns of paw placement were also disrupted in *pcd*. While the average distance between hind paw placement and previous forepaw position was the same in *pcd* and controls ($F_{(14,1)}$ = 0.44, p = 0.52), it was more variable in *pcd* ($F_{(168,1)}$ = 75.30, p < 0.001; *Figure 4G*), across speeds.

Taken together, these results reveal that both spatial and temporal measures of interlimb coordination during overground locomotion were altered in *pcd* mice.

## Whole-body coordination deficits in *pcd*

Movements not just of the limbs, but of the entire body need to be coordinated during locomotion. In order to characterize whole-body locomotor coordination in both control and *pcd* mice, we analyzed their head and tail movements while they walked freely across the corridor (*Figure 5*; *Figure 5—figure supplement 1*).

In control mice, lateral movements of the nose and tail were small (*Figure 5A,B*, black). In striking contrast, however, both the nose and the tail of *pcd* mice exhibited large side-to-side oscillations during the locomotor cycle (*Figure 5A,B*, purple; *Video 2*) (compared to size-matched controls: nose $F_{(13.48,1)}$ = 5.49, p < 0.05, tail $F_{(415,1)}$ = 91.01, p < 0.001).

Not just the amplitude, but also the timing of nose and tail movements relative to paw stride cycles was altered in *pcd*. In control mice, both the tail and nose were phase-locked to the stride cycle across walking speeds (*Figure 5D*). However, in *pcd* both nose and tail (*Figure 5E*) became increasingly phase-lagged at faster speeds (tail: $F_{(401.02,1)}$ = 5.55,p <0 0.05; nose: $F_{(53.61,1)}$ = 4.89,p <0 0.05 for speeds above 0.1 m/s). Interestingly, the phase relationships for both the nose and the tail in *pcd* were consistent with their being time-, rather than phase-locked, to the locomotor cycle (*Figure 5—figure supplement 1*).

The large, sinusoidal amplitude of the tail oscillations and their time-delayed relationship to hind paw movement suggested to us that tail movements in *pcd* could reflect passive consequences of the forward movement of the hind limbs. To investigate this possibility, we generated a simple geometrical model of a mouse (*Figure 5C*; 'Materials and methods') in which the tail was anchored at a right angle to a point that bisected a line connecting the two hind paws. We used the data from *Figures 3 and 4* to model the hind paw alternation and calculated the predicted side-to-side tail trajectories across walking speeds. As illustrated in *Video 3*, the orthogonal coupling between the line connecting the hind paws and the tail segments caused the modeled tail to oscillate from side-to-side as the hind paws moved forward.

The side-to-side tail movements predicted by the passive model were strikingly similar to the tail oscillations of *pcd* mice (compare grey dashed lines with purple traces in *Figure 5A,B*). As shown in *Figure 5F*, the tail oscillations of *pcd* mice were fit very well, across walking speeds and tail segments, with a passive model that incorporated a 31 ms time delay for the base of the tail (reflecting the initial inertia of the mouse's rear) plus a fixed delay per additional tail segment (see 'Materials and methods'). A similar model for the nose (*Figure 5*) was also consistent with a time-delayed (96 ms) side-to-side nose movement in *pcd*. Importantly, these oscillations cannot be accounted for by differences in hind limbs between control and *pcd*, because: (1) hindpaw alternation is unaltered in

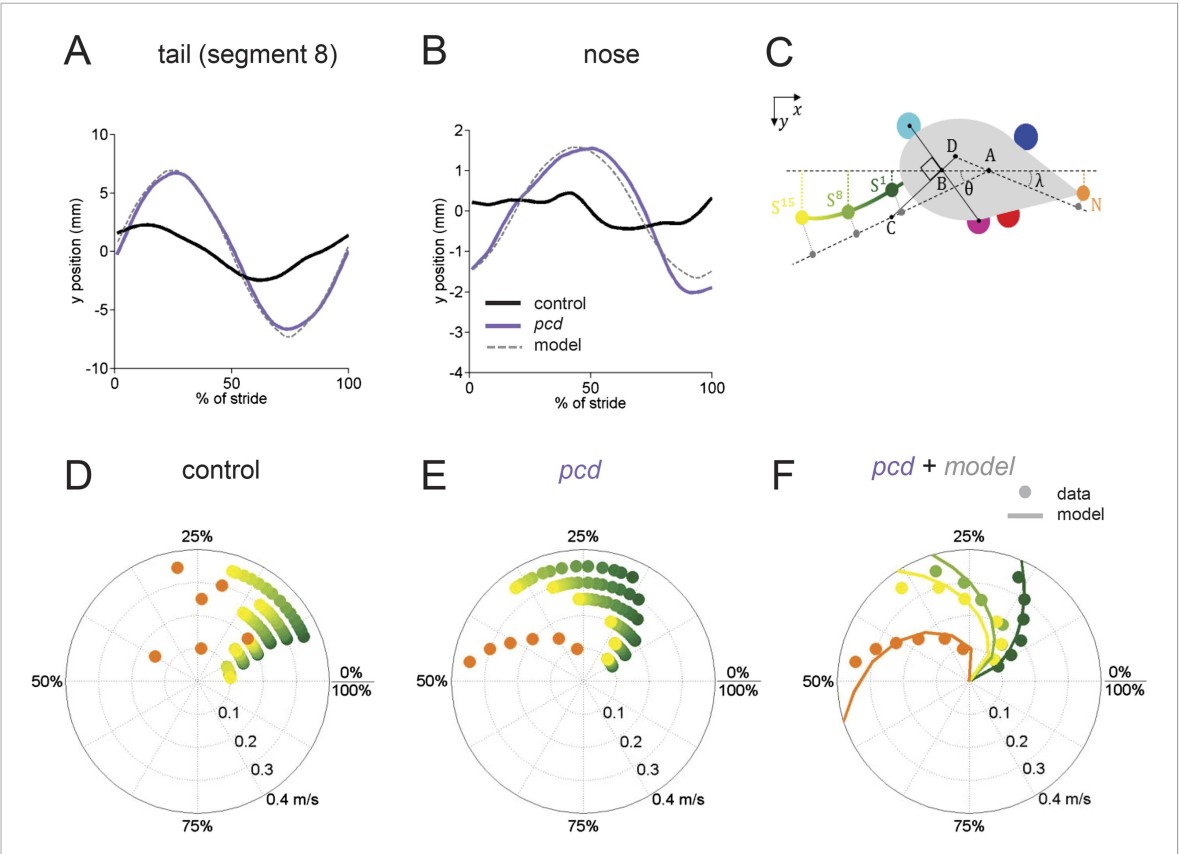

**Figure 5**. The tail and nose movements of pcd mice can be modeled as a passive consequence of the forward motion of the hind limbs. (**A**, **B**) Averaged lateral trajectory of tail segment 8 (**A**) and nose (**B**), relative to the mid-point between the hind paws, for animals walking at 0.25–0.30 m/s, for size-matched controls (black), pcds (purple) and model (dashed gray). (**C**) Geometric model of the tail and nose. The lateral position of each tail segment at every time step is given by $S_{yi} = \_AS_i sin\theta$, whereas the lateral position of the nose is given by $N_y = \_DN \_sin\lambda$. (**D**, **E**) Phase of maximum correlation between the forward position of the hind paws and the lateral trajectories of each tail segment (green-yellow gradient) and the nose (orange). (**F**) The maximum correlation phases of the passive geometric model (lines) are superimposed on the observed values (circles) for tail segments 1 (dark green), 8 (light green), 15 (yellow), and nose (orange).

The following figure supplements are available for figure 5:

**Figure supplement 1**. Nose and tail movements across speed bins.

**Figure supplement 2**. Results summary for young pcd and size-matched controls.

pcd (**Figure 4A,B**); (2) hindlimb double support is unaltered in pcd (**Figure 4F**); and (3) though the base of support is wider in *pcd*, the model predicts that this difference alone would result in smaller, not larger tail oscillations in *pcd*. Thus, in *pcd*, the tail and nose appear to move as a passive consequence of forward limb motion. Further, our results suggest that this movement must be actively canceled in wildtype mice to keep the body axis aligned for forward movement.

To visualize and quantify impairments in whole-body coordination, we compared vertical (z) trajectories for each body part, normalized to 100% of the stride cycle. **Figure 6** summarizes the trajectories of individual body parts as well as interlimb and whole body coordination of speed- and size-matched control and *pcd* mice. During locomotion in control mice, the movement of different parts of the body is synchronized, and vertical nose and tail movements are relatively small (**Figure 6A**, left). In *pcd*, however, spatial and temporal coordination across the body is dramatically impaired (**Figure 6A**, right). Finally, a correlation analysis reveals that in control mice, the movement of most body parts is either strongly correlated or anti-correlated (i.e. they move either in–or out-of-phase with each other; **Figure 6B**, left, red and blue). In *pcd* mice, however, the correlations are

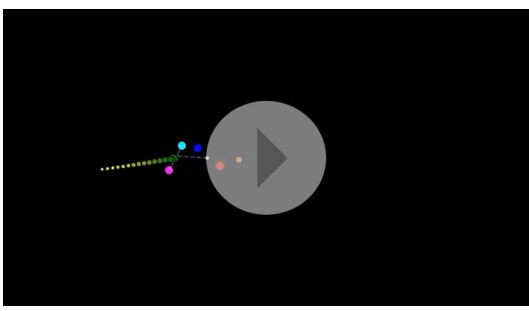

**Video 3.** Passive nose and tail model. Passive model of nose and tail for a mouse walking at 0.2 m/s. The forward movements of the paws were modeled according to the data described in *Figures 3 and 4*. The lateral movements of the nose and tail segments are predicted from a model in which an orthogonal projection transforms the forward movements of the hind paws (solid white line) into lateral movements of the tail (dashed white line) and nose with a fixed time-delay for each element. As a result of this orthogonal coupling, the nose and tail oscillate laterally as a passive consequence of the forward motion of the hind paws.

weaker and more variable (*Figure 6B*, right). This lack of correlational structure reflects the failure of *pcd* mice to synchronize the movements of different parts of the body.

Taken together, the results of *Figures 3–6* suggest that while the forward motion of individual paws is largely spared, ataxic *pcd* mice have specific deficits in coordinating movement in three dimensions across joints, limbs, and body.

## Discussion

Establishing appropriate, sensitive, and specific behavioral measures is an essential first step for investigating relationships between brain and behavior (*Clark et al., 2013*; *Anderson and Perona, 2014*). Although deficits in locomotor coordination are readily visible to the human eye, identifying the specific, quantitative features of gait ataxia has been more difficult (*Leblond et al., 2003*; *Cendelin et al., 2010*; *Krug et al., 2013*). Here we used the custom-built Loco-Mouse system to analyze locomotor kinematics and coordination in freely walking mice. The high spatiotemporal resolution and throughput of this system provides the most comprehensive description of locomotor kinematics in freely walking mice to-date and allowed us to develop a novel analysis framework for mouse locomotion that revealed fundamental features of gait ataxia. Our main findings are (1) Basic paw stride parameters can be predicted solely based on walking speed and body size. (2) This relationship holds in visibly ataxic *pcd* mutants, indicating that changes in these stride parameters in ataxic mice do not reflect fundamental features of ataxia. (3) While the forward motion of the paws is spared in *pcd*, coordination across joints, limbs, and the body is selectively impaired. (4) The nose and tail of *pcd* mice oscillate as a passive consequence of forward hindlimb motion. Taken together, this pattern of deficits reveals that gait ataxia in *pcd* mice involves specific impairments in locomotor coordination across joints, limbs, and body parts.

### LocoMouse

Because of the significant challenges associated with quantifying whole-body coordination in freely walking animals, assessments of mouse motor coordination phenotypes often rely on indirect measures (*Mullen et al., 1976*; *Herbin et al., 2007*; *Lalonde and Strazielle, 2007*; *Guillot et al., 2008*; *Brooks and Dunnett, 2009*; *Cendelin et al., 2010*; *Stroobants et al., 2012*; *Sheets et al., 2013*; *Suidan et al., 2013*; *Camera et al., 2014*), such as time to fall from a rotarod (*Walter et al., 2006*; *Lalonde and Strazielle, 2007*) or a fixed bar (*Kim et al., 2009*; *Cendelin, 2014*), or mis-steps on a ladder (*Vinueza Veloz et al., 2014*). While these can be sensitive markers for global motor dysfunction, they lack specificity. Moreover, performance on coordination tasks, or even on a treadmill (*Hamers et al., 2001*; *Hoogland et al., 2015*), does not necessarily correspond to the degree of gait ataxia during overground locomotion (*Herbin et al., 2007*; *Lalonde and Strazielle, 2007*; *Guillot et al., 2008*; *Cendelin et al., 2010*; *Stroobants et al., 2012*; *Suidan et al., 2013*; *Camera et al., 2014*).

Our goal in developing LocoMouse was to create a system that would be as useful for assessing cerebellar contributions to locomotor coordination as existing systems have been for analyzing spinal cord control of locomotor pattern generation (e.g. *Crone et al., 2009*; *Bellardita and Kiehn, 2015*). LocoMouse presents several advantages when compared to available systems for analyzing mouse locomotion (*Hamers et al., 2001*; *Hoogland et al., 2015*; *Leblond et al., 2003*; *Kale et al., 2004*; *Garnier et al., 2008*; *Zörner et al., 2010*). Mice walk freely and naturally across the corridor and throughput is maximized via fully automated data collection and analysis. The high spatiotemporal

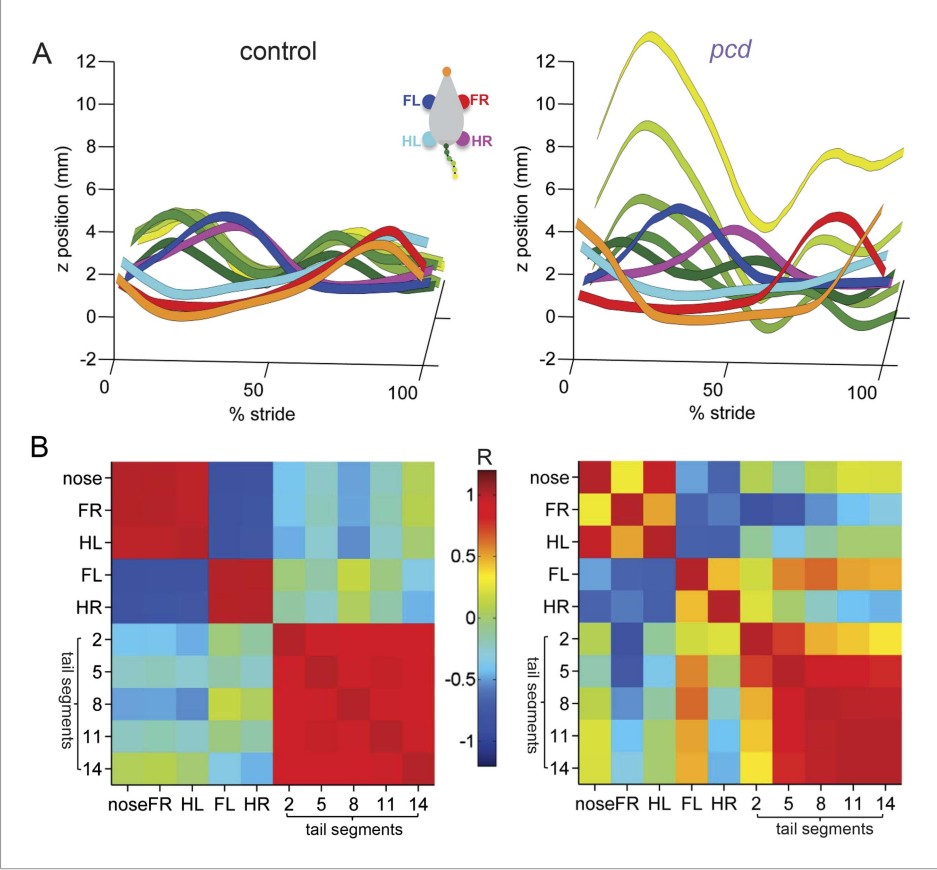

**Figure 6**. Visualization of impaired whole-body coordination in *pcd*. (**A**) Ribbon plots showing average vertical (z) trajectories for nose, paw and select tail (2, 5, 8, 11, 14) segments for size-matched control (Left, N = 11)) and *pcd* (Right, N = 3) mice walking at 0.20–0.25 m/s. Data are presented relative to 100% of the stride cycle of the FR paw (x-axis). Nose and paw trajectories are z position relative to floor; tail is z relative to floor with mean vertical position of the of base of the tail subtracted for clarity. (**B**) Matrix of correlation coefficients computed for average vertical trajectories of control (Left) and *pcd* (Right). Color bar is value of correlation coefficient.

resolution provides detailed 3D paw kinematics. LocoMouse also tracks nose and tail movements during locomotion, which have not been previously reported in freely walking mice. While the automated tracking algorithm does not measure joint angles, these can be incorporated either through hand-labeling or potentially with the use of additional markers.

We exploited the large, multidimensional dataset generated by LocoMouse to establish a quantitative framework for analyzing locomotor coordination in freely walking mice. There were two key features of our approach that allowed us to isolate specific deficits of ataxia. The first was quantitatively accounting for variability across strides and mice. Although individual strides at first appeared quite variable across our diverse set of wildtype mice, we found that wildtype paw kinematics were readily and quantitatively predictable based on walking speed and body size, even down to the level of 3D trajectories (*Figure 2*, *Figure 2—figure supplement 1*). Second, we placed particular emphasis on quantifying coordination across the body.

## A quantitative framework for locomotor coordination reveals fundamental features of mouse gait ataxia

The statistical models we developed based on a library of data from wildtype mice (*Figure 2*, *Figure 2—figure supplement 1*) accurately predicted the forward paw motion of ataxic *pcd* mice (*Figure 3*). This indicates that observed differences in basic stride parameters were a secondary consequence of differences in body size and walking speed. Further, the forward motion of individual

paws was indistinguishable from that of size- and speed-matched controls, down to the level of detailed paw trajectories. This important result highlights that failure to account for differences in walking speed and body size when comparing data across mice and strides can lead to nonspecific effects being misinterpreted as symptoms of ataxia (*Koopmans et al., 2007*; *Cendelin et al., 2010*; *Wuehr et al., 2012*; *Batka et al., 2014*). Moreover, it is likely that by focusing primarily on the forward movement of individual paws, many existing analyses fail to capture the fundamental features of ataxia.

While differences in forward paw motion could be fully accounted for by differences in walking speed and body size, in contrast, off-axis paw trajectories, interlimb, and whole-body coordination revealed specific patterns of impairment in *pcd* (*Figures 4–6*). Differences in off-axis movements (*Figure 3*, *Figure 3—figure supplement 1*) suggest that *pcd* mice, like human cerebellar patients, are unable to coordinate movements across joints within the limb to perform normal strides in 3D (*Earhart and Bastian, 2001*; *Bastian et al., 1996*). Further, *pcd* mice exhibited impaired spatial and temporal coordination of movements across the four limbs, nose, and tail. Interestingly, while front-hind paw coupling was dramatically altered in *pcd*, left-right alternation was preserved entirely, consistent with the idea that such alternation is generated within the spinal cord itself (*Crone et al., 2009*; *Kiehn, 2011*; *Dougherty et al., 2013*). Finally, the large, oscillatory nose and tail movements observed in *pcd* were not just random, but were successfully modeled as a failure to predict and compensate for the passive consequences of forward motion of the hind limbs.

## Cerebellar contributions to coordinated movement

The major neuroanatomical finding in *pcd* is the complete postnatal degeneration of cerebellar Purkinje cells and subsequent loss of granule cells and related structures (*Mullen et al., 1976*; *Lalonde and Strazielle, 2007*; *Morton and Bastian, 2007*). In light of these extensive anatomical defects and the existing body of literature on mouse ataxia, the remarkable preservation of forward paw motion in *pcd* (*Figure 3E*) was surprising. Moreover, previous studies have associated Purkinje cell modulation with the step cycle of individual limbs (*Armstrong and Edgley, 1984*; *Edgley and Lidierth, 1988*; *Udo et al., 2004*) and movement kinematics (*Pasalar et al., 2006*; *Heiney et al., 2014*). The most likely interpretation of the surprisingly intact forward paw motion in *pcd* is that it reflects the presence of inevitable compensatory mechanisms resulting from the chronic loss of Purkinje cells. Given this capacity for compensation, the specific and persistent impairments in multi-joint, interlimb, and whole-body coordination in *pcd* are particularly striking.

While many lines of evidence suggest that the cerebellum provides internal models for motor control (*Wolpert et al., 1998*; *Ito, 2008*), there has been disagreement about the nature of these models (*Ebner and Pasalar, 2008*; *Medina, 2011*). Studies from some systems, including eye movements, have suggested that the cerebellum acts as an inverse model that computes a command to achieve desired movement (*Shidara et al., 1993*). Other work, particularly on reaching movements, the cerebellar-like nuclei of electric fish, and locomotor adaptation, has suggested that the cerebellum provides a forward model that predicts the consequences of movements (*Bastian et al., 1996*; *Kennedy et al., 2014*) (*Pasalar et al., 2006*) (*Morton and Bastian, 2006*). These predictions can then be used to optimize joint angle combinations within a limb (*Bastian et al., 1996*), synchronize interlimb coordination (*Figure 4*), or to cancel out the unintended passive consequences of movements of other parts of the body (*Figure 5*). We found that while on-axis paw kinematics are preserved in the absence of cerebellar cortical output, the ability to predict and actively cancel the passive consequences of movements of other parts of the body appears to be beyond the limits of compensatory mechanisms available to *pcd* mice. Therefore our results raise the intriguing possibility that the absence of forward models, rather than a failure to execute appropriate movement kinematics per se, could form the basis for impaired coordination associated with gait ataxia in mice.

Although cerebellar involvement is a shared feature across ataxic mouse mutants, the details of the motor phenotypes are different for different mutations (*Lalonde and Strazielle, 2007*). Similarly, patients with cerebellar damage exhibit varying degrees and features of ataxia (*Morton and Bastian, 2003, 2007*; *Yu et al., 2014*). Given the diversity of cerebellar phenotypes, it is likely that the specific features of gait ataxia will vary across mouse models. The novel quantitative framework for mouse locomotion presented here highlights the importance of considering 3D, interlimb and whole-body coordination and dissociating them from the control of individual paw kinematics, particularly when

analyzing cerebellar contributions to locomotion. Together with the sophisticated genetic tools available for manipulating neural circuits in mice, the current approach makes mouse locomotion a powerful system for investigating the neural control of coordinated movement and establishing relationships between neural circuit activity and behavior.

## Materials and methods

### Animals

All procedures were reviewed and performed in accordance with the Champalimaud Centre for the Unknown Ethics Committee guidelines, and approved by the Portuguese Direcção Geral de Veterinária (Ref. No. 0421/000/000/2015).

C57BL/6 mice were housed in institutional standard cages (3 animals per cage) on a reversed 12-hr light/12-hr dark cycle with *ad libitum* access to water and food. Heterozygous *Purkinje cell degeneration* mice on a C57BL/6 background were obtained from Jackson labs (#0537 B6.BR-Agtpbp1*pcd*/J).

Experiments were conducted in two groups: (a) wildtype controls (n = 9602; N = 34 mice; 23 male; 11 female; 7-33g, 30–114 days old) and (b) homozygous *pcd* mice (n = 3052; N = 3 mice; 2 female, 1 male; 10–16 g; run at several ages each between 41-154 days old) and their littermates (n = 2256; N = 7 mice; 3 female, 4 male; 15–40 g; 34–190 days old). Size-matched controls for *pcd* animals (n = 3400; N = 11 mice) were taken from the wildtype data set (*Figure 2—figure supplement 1* and *Figure 3—figure supplement 1*).

Because of extra-cerebellar degeneration in pcd mice after postnatal day 50, we performed a separate analysis to verify that our main findings held in the youngest pcd mice (*Figure 5—figure supplement 2*).

### LocoMouse setup

A custom-designed setup was developed to assess whole body coordination during overground locomotion in mice (*Figure 1*). The LocoMouse apparatus consists of a clear glass corridor, 66.5 cm long, 4.5 cm wide and 20 cm high. Mice were filmed crossing the corridor with a high-resolution, high-speed camera (Bonito CL-400B, Allied Vision Technologies, https://www.alliedvision.com). A mirror (66 cm × 16 cm) was placed below the corridor at an angle of ~45° to allow simultaneous collection of side and bottom views in order to generate three-dimensional tracking data. Lighting consisted of a matrix of LEDs that emitted cool white light positioned to maximize contrast and reduce reflection. Infrared sensors positioned along the runway automatically triggered the camera and acquisition software once the mouse entered the corridor and stopped the acquisition once the mouse reached the other end of the corridor or after 25 s.

### Data collection

Mice were handled by the experimenter and allowed to acclimate in the LocoMouse setup for several minutes on multiple occasions before data collection. Animals were weighed before each session. Mice walked freely between two dark boxes on either end of the glass corridor. The automatic triggering system was critical for allowing mice to self-initiate trials, which reduced animal stress without compromising the quantity of data collected. No food or water restriction or reward was used.

10-15 corridor passages (trials) were collected in each of five daily sessions. A total of 36,369 strides were collected from wildtype controls, which corresponds to 1069 ± 266 strides per mouse and 267 ± 66 strides per paw. For *pcd* and littermates we collected an average of 4252 ± 778 strides per *pcd* mouse (1063 ± 32 strides per animal per paw) and 1310 ± 934 strides per littermate mouse (328 ± 11 strides per animal per paw). Animals were not required to walk continuously throughout a trial; if the animal stopped mid-trial, the data before and after the halt were still analyzed.

### Data acquisition

Movies were collected at 400 frames per second with a spatial resolution of 1440x250 pixels. Acquisition software was written in Labview and uses 2 National Instruments boards (PCIe 1433 and BNC 2120) to record and save the movies, in real time. The tracking algorithm and data analysis software were written in Matlab (Mathworks) and performed offline. The LocoMouse Tracker code used in this paper can be downloaded from GitHub (https://github.com/careylab/LocoMouse).

## Tracking algorithm

### Overview

To maximize throughput we used a computer vision algorithm to allow automatic, markerless tracking of features of interest (without the need for surface markers or manual initialization of feature tracks) (*Figure 1* and *Figure 1—figure supplement 1*). The algorithm's output was the set of 3D coordinates of the features of interest, which were: the center of each of the four paws, the snout, and the tail divided into 15 points, for each frame of the movies. The output of the tracking system was visually inspected for each trial. Fewer than 10% of the trials were excluded due to tracking failure (typically due to exploration or grooming behavior that resulted in erroneous swing and stance point detection). Further validation of the automated tracking performance is shown in *Figure 1—figure supplement 1*.

We used hand labeled data to train linear Support Vector Machine (SVM) classifiers for each feature and each view (side and bottom) independently. Positive examples were hand labeled on a set of 81 images from a single training video while 10 negative examples for each feature were randomly picked from the same images. Training the 6 SVM classifiers took approximately 1 hr on a machine with an Intel Core i7-3770 CPU and 16 GB or RAM. All experiments were performed with these SVM classifiers with no need to retrain on additional data. For each trial, the candidate locations for each feature were obtained by filtering the input images with the trained detectors. For the paws and snout, the resulting image regions were clustered into a small number of point locations using a standard non-maxima suppression algorithm. Once the candidate features were identified by the SVM, we considered the temporal trajectories and selected the candidate tracks that maximized per image detection scores while minimizing frame-by-frame displacement with a method based on existing multi-target tracking frameworks (*Russell et al., 2011*) that formulate tracking as the maximum a posteriori probability estimation over a Bayesian network of candidate locations on each image. Best bottom view tracks (x and y) were computed first and then the best side view track (z, accounting for image distortion from the optics) that matched the bottom view track was selected. For the tail, the interest regions were matched across views and the largest resulting 3D region was picked as the final detection. The tail was detected independently for every image. Following computation of feature tracks, pixel values were converted to millimeters for further analysis.

### Detailed description

The LocoMouse Tracker code developed and used in this paper has been deposited at GitHub (https://github.com/careylab/LocoMouse) and updated versions are available through our website. The LocoMouse tracker was developed in MATLAB R2013b (The MathWorks). Some auxiliary packages can be found at the Matlab Central File Exchange (http://www.mathworks.com/matlabcentral/fileexchange/). The method also relies on the LIBSVM library (http://www.csie.ntu.edu.tw/~cjlin/libsvm/) for Support Vector Machines and the code from (*Russell et al., 2011*) for Multi-target tracking.

### Removing background and correcting camera/mirror distortion

An image of the corridor was recorded before every session. To remove the background we subtracted this image from every frame recorded by the system. To correct for image distortion we recorded a video of a white spherical object moving along the full volume of the glass corridor. The two projections of the object were detected by thresholding the (grayscale) image at 80% (after removing the background). The horizontal line splitting the image into the bottom and side views was defined manually at this point. The corrective image transformation makes the bottom and side projections of the object match along the vertical line and was computed via least squares. The corrective transformation depends on the position of the camera relative to the setup and was performed once for every configuration. Unless stated otherwise, all steps of the algorithm are performed on corrected images.

### Computing bounding box around mouse

We further isolated the mouse by computing a tight bounding box around it. We started by thresholding every input image at 10% and using a median filter to remove noise. For every image, the edges of the box are defined as the first and last rows and columns to have white pixels. To account for occlusions and noise, the final size of the bounding box was defined as the smaller of the mean size

plus three standard deviations, and the maximum observed size. The final trajectory of the bounding box over the video was determined by filtering the box position calculated for each image with a moving average filter of width 5. All further steps in detecting the interest features were performed within the bounding box.

## Training the detectors

Feature detectors for the paws, snout and tail were trained using the LIBSVM library. Positive examples were manually annotated on a set of 81 training images from a single movie. The size of the detectors was chosen manually such that the feature lied within the detector box. Each view was treated independently, unless otherwise stated. The sizes of the different feature detectors can be found in *Table 1*.

The system assumes mice move from left to right. Images were flipped horizontally when otherwise. For every feature, 10 negative examples were randomly extracted from within the bounding box of the mice for each of the 81 training images (excluding the positively labelled regions).

## Detecting paw and snout candidates

The outputs of LIBSVM were used to filter the pre-processed input images. These filters provided a score representing the likelihood of a pixel being part of each feature (paw or snout). These per-pixel scores were reduced into a small number of candidate locations with a standard Non-Maximum Suppression (NMS) algorithm (http://vision.ucsd.edu/~pdollar/toolbox/doc/) which clusters positively classified pixels into local maxima. Since on the side view there is considerable overlap between the features, we used a more conservative version of the NMS algorithm, which results in more candidate locations.

## Combining bottom and side view candidates

The 2D candidate locations from each view were combined into 3D candidates by matching their coordinates on the shared axis (horizontal axis). As exact matches do not occur, we used a tolerance of 30% of the feature's detector size along the horizontal axis for matching.

We allowed each bottom view candidate to be matched to many side view candidates, but each side view candidate was only (generally) matched to a single bottom view candidate. In ambiguous configurations, additional information about the velocity of the candidates was used to find the best match. Candidates were classified as moving/not moving according to the pixel count within the feature box size after subtracting the previous input image to the current image. When candidates could not be disambiguated, both options were allowed to remain.

## Computing feature tracks

We used location priors to distinguish between instances of the same class (e.g., to identify which paw was which). Based on the configuration of the corridor, the priors were defined as the inverse of the distance to the closest corner (i.e. on the bottom view, the front right paw is closer to the bottom right corner while the hind left paw is closer to the top left corner). The prior was defined on a normalized box which allows it to be fit to a bounding box of any size. Locations on the normalized box with a distance greater than 0.6 had their prior score set to zero.

## Multi-target tracking

After processing the candidate locations at each image, tracking on the bottom view was performed using the multi-target tracking algorithm of (*Russell et al., 2011*). This algorithm finds the tracks over all the images which maximizes the per image detection score (weighted by the location prior) while

**Table 1**. Image feature size

| Feature | Bottom view size (in pixels) | Side view size (in pixels) |
| --- | --- | --- |
| Paw | 30 x 30 | 20 x 30 |
| Snout | 40 x 40 | 20 x 40 |
| Tail segments | 30 x 30 | 25 x 30 |

minimizing the in-between image distance. In practice the inverse distance was considered and the problem is formulated as

$$\max_{\boldsymbol{x} \in L^{N,F}} C(\boldsymbol{x}) = \sum_{o=1}^{N} \left[ \sum_{t=1}^{F} U\left(x_{o,t}\right) + \alpha \sum_{t=1}^{F-1} P\left(x_{o,t}, x_{o,t+1}\right) \right],$$

where $\boldsymbol{x}$ is a possible (multi-target) track, $L$ is the number of possible locations, $N$ is the number of objects to track, $F$ the number of images in the video, $x_{o,t}$ a candidate location for object $o$ at time $t$, $U(.)$ the image detection score weighted by the location prior, $P(.,.)$ the inverse of the image distance between two points, and $\alpha$ the relative weight between image detection scores and image transitions (set empirically to 0.1). In practice we discarded all transitions that exceeded v pixels between two images, where v is the sum of the (variable) displacement of the bounding box and a fixed value of 15 pixels. An additional constraint was added such that only one object could occupy the same candidate location at the same instance in time.

For the side view, we used the same approach to find the most likely Z trajectory given the already computed X and Y tracks.

## Data analysis

The stride cycles of individual paws were automatically broken down into swing and stance phases based on the first derivative of the paw position trajectories. Individual strides were defined from stance onset to subsequent stance onset. For each stride, average *walking speed* was calculated by dividing the forward motion of the body center during that stride by the stride duration. All data was sorted into speed bins (0.05 m/s bin width) in a stridewise manner, with a minimum stride count criterion of 5 strides per bin, per animal. Individual limb movements and interlimb coordination were calculated as follows:

## Individual limb parameters

### Cadence

Inverse of stride duration

### Swing velocity

x displacement of single limb during swing phase divided by swing duration.

### Stride length

x displacement from touchdown to touchdown of single limb.

### Stance duration

Time in milliseconds that foot is on the ground during stride.

### Trajectories

Trajectories were aligned to swing onset and resampled to 100 equidistant points using linear interpolation. Interpolated trajectories were then binned by speed and the average trajectory was computed for each individual animal and smoothed with a Savitzky-Golay first-order filter with a 3 point window size.

## Interlimb and whole-body coordination parameters

### Stance phase

relative timing of limb touchdowns to stride cycle of reference paw. Calculated as: (stance time −stance time_{reference paw})/stride duration.

### Supports

Support types were categorized by number of paws on ground expressed as a percentage of the total stride duration for each stride. Paw support categories are four, three, two diagonal, two other (homolateral and homologous), one, and zero.

### Double support

Double support for each limb is defined as the percentage of the stride cycle between the touch down of a reference paw to lift-off of the contralateral paw. Because at higher speeds (running), the opposing limb lifts off before the reference paw touches down, we included negative double support by looking backwards in time by 25% of the stride cycle duration. Positive values of double support indicate that contralateral lift-off occurred after reference paw touch down, and negative values indicate that contralateral lift-off occurred before reference paw touch down.

### Paw distance

The x,y distance from where the front paw lifted off to where the ipsilateral hind paw touched down on the subsequent stride.

### Tail and nose phases

For each speed bin we correlate the stridewise tail and nose trajectories with the trajectory given by the difference between the forward position of the hind right paw and the forward position of the hind left paw (also normalized to the stride). The phase is then calculated by the delay in which this correlation is maximized.

### Correlation matrices

Correlation coefficients were computed for average z trajectories normalized to 100% of the FR stride cycle.

## Multi-level linear mixed-effects models and statistical analysis

Because of the complex nature of the data set, including nested data with varying number of trials per animal and data points per speed bin, data analysis and statistical comparisons were performed with linear mixed effects models.

## Modeling stride parameters across control mice

We used linear mixed effects models (*Bates et al., 2013*) to analyze our data and quantify the relationship between fixed (speed, gender, weight, length, age) and random (subject) effects and values for each stride parameter.

To linearize data for inclusion in the model, we first plotted the values of each stride parameter vs walking speed and generated fits to the data. Different fitting curves (linear, quadratic, cubic, inverse, logarithmic, exponential, power) were tested and selected based on the distribution of residuals and $R^2$ values. The curves that provided the best fits to each parameter are given in *Figure 2—figure supplement 1*.

After transforming the data according to these best fit curves, we next asked to what extent the fixed and random effects contributed to remaining variability in the values of each parameter. For the fixed terms, we tested different equations, using additive and interaction properties. Random terms took into consideration differences in both slopes and intercepts. The equations were selected based on the following criteria: $R^2$ values (marginal and conditional, (*Nakagawa and Schielzeth, 2012*), likelihood ratio tests (comparing goodness of fit across equations) and collinearity of effects (*Figure 2—figure supplement 1*). Due to collinearity, in many cases body length (measured directly from the movies) and weight provided good fits to the data. We chose to use weight throughout because it provided a platform-independent metric for body size that should be more reproducible.

## Statistical analyses

Statistical analyses were done in Matlab and R. For all comparisons, models were selected by comparing equations specifying additive fixed-effects terms with those specifying n-way interaction terms using a likelihood-ratio test and inspection of statistical significance of included terms. Depending on the comparison, fixed-effects terms included a subset of the following variables: speed, genotype and paw. All models were random-intercepts models with subject as a random covariate. Unless otherwise indicated, results are reported as conditional F tests with Satterthwaite degrees of freedom correction. All variability analyses were based on coefficients of variation (CV).

## Geometric model of the tail and nose

The simulations of the lateral movements of the tail and nose were carried out in three steps. (1) First, we estimated the desired stride parameters for each hind paw. The temporal parameters of the stride

(swing, stance and stride durations) were taken from *Figure 2*, calculated for an animal of 15g (small animal control). We calculated the stride lengths by estimating the final desired position of the paw (relative to the nose of the animal) at the end of the stride; this value was also taken from small animal control data. (2) We calculated the forward trajectory for each paw using a constant swing velocity model given by the stride length divided by the swing duration. We then calculated the periodic forward oscillations of the two hind paws by calculating the difference between their forward trajectories. The phasing of hind paw alternation was taken from the small animal control data (*Figure 4A*). (3) Finally, we converted the forward movements of the paws into lateral movements of the tail and nose. We calculated tail and nose trajectories that would be predicted from a purely passive coupling of forward motion of the hind paws with lateral tail/nose motion through the geometric relations $\overrightarrow{BC} \perp \overrightarrow{LR}$ and $\overrightarrow{BD} \perp \overrightarrow{LR}$ (*Figure 5A*). We bisected the line segment that connects the two hind paws, with an orthogonal line segment (CD). For each tail segment $S_i$, at each timestep t, we define a vector with constant length, originating in A, along the direction $\overrightarrow{AC}$ (gray points along the line AC). The lateral position of each segment of the tail was then given by $S_{yi} = L_i \sin \theta$, where $L_i$ is the distance between the center of tail oscillation (A) and segment i, and $\theta$ is the angle between segment (AC) and the anterior–posterior axis of the animal. Similarly, for the nose, we defined a vector with constant length, originating in A, along the direction $\overrightarrow{DA}$. The position of the nose was given by $N_y = L \sin \lambda$, where L is the distance between the center of nose oscillation (A) and the nose, and $\lambda$ is the angle between segment (DA) and the anterior–posterior axis of the animal. The final position of each tail segment $S^i$ and nose $N$ were then given by each of these vectors delayed by a fixed amount. The time delays for the nose and base of the tail were estimated by linear regression on a plot of phase as a function of stride frequency (cadence) (*Figure 5—figure supplement 1H*). Delays between subsequent tail segments decreased according to the equation delay = −0.23 *i + 3.97, where i is the segment number, starting at the base of the tail.

## Acknowledgements

We thank Carla Matos for her valuable contributions to the early phases of this project, visiting students J Pascoal, V Jayaram, and I Prata for technical assistance, and S Lisberger, S Sober, and members of the Carey lab and the Champalimaud Neuroscience Program for useful discussion and comments on a previous version of the manuscript. This work was supported by an International Early Career Scientist grant from the Howard Hughes Medical Institute, a Starting Grant from the European Research Council (both to MRC), and predoctoral fellowships from the Portuguese Foundation for Science and Technology to ASM and DMD.

## Additional information

### Funding

| Funder | Grant reference | Author |
| --- | --- | --- |
| European Research Council (ERC) | Starting Grant | Megan R Carey |
| Howard Hughes Medical Institute (HHMI) | International Early Career Scientist | Megan R Carey |
| Fundação para a Ciência e a Tecnologia (FCT) | Graduate Student Fellowship | Dana M Darmohray, Ana S Machado |

The funders had no role in study design, data collection and interpretation, or the decision to submit the work for publication.

### Author contributions

ASM, Conception and design, Acquisition of data, Analysis and interpretation of data, Drafting or revising the article; DMD, MRC, Conception and design, Analysis and interpretation of data, Drafting or revising the article; JF, Developed tracking algorithm, Conception and design, Analysis and interpretation of data, Drafting or revising the article; HGM, Analysis and interpretation of data, Drafting or revising the article

**Author ORCIDs**

Megan R Carey, http://orcid.org/0000-0002-4499-1657

**Ethics**

Animal experimentation: All procedures were reviewed and performed in accordance with the Champalimaud Centre for the Unknown Ethics Committee guidelines, and approved by the Portuguese Direcção Geral de Veterinária (Ref. No. 0421/000/000/2015).

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
