## [Decision Letter]

[Editors’ note: this article was originally rejected after discussions between the reviewers, but the authors were invited to resubmit after an appeal against the decision.]

Thank you for choosing to send your work entitled "A quantitative framework for whole-body coordination reveals specific deficits in freely walking ataxic mice" for consideration at *eLife*. Your full submission has been evaluated by Eve Marder (Senior Editor) and three peer reviewers, one of whom is a member of our Board of Reviewing Editors, and the decision was reached after discussions between the reviewers. Based on our discussions and the individual reviews below, we regret to inform you that your work will not be considered further for publication in *eLife*.

All the reviewers find that the study is very interesting and that the method presented for monitoring locomotion in relation to the cerebellar motor function and other motor functions is potentially strong. There are, however, concerns whether the automation procedure is making correct assessments of the locomotor parameters, that gaits are not taken into accounts in the description, that there is a lack of statistical evaluation of intra-limb coordination, and that the differences in full body movements may not account completely for the phenotype. To meet these concerns would require substantial reanalysis of the data and reconsidering the main conclusions of the study. Even with a new analysis there is a feeling that the results presented will mainly be a technical advance rather than a conceptual advance for cerebellar motor function or the locomotor field at large. The reasons for this are that there already are cerebellar specific mutants that have been analyzed in similar ways and that the locomotor data in wild-type are confirmative and lacking gait parameters. For these reasons the manuscript as presently configured is not suitable as a regular paper in *eLife*. The details of the concerns and suggestions for improvements are detailed in the attached review reports.

*eLife* has "Tools and Resources" that present methods papers. If you feel that you will be able to reanalyze the data as outlined and make a case that this method is better and more usable than other methods already published it might be an option for you to consider submitting your work in this form to *eLife*.

*Reviewer #1*:

The paper of Machado et al. introduces a camera-based motion capture system for the analysis of walking movements in the mouse. The heart of the system is machine learning operating on resampled images of high speed cameras for mice walking in a glass corridor. To illustrate its applicability, the paper compares the walking movements of wildtype and Purkinje cell degeneration (*pcd*) mice.

In general, for a paper introducing a novel analysis tool, I think it must be assessed whether the method provides measurements of highly useful parameters, with high explanatory value, that goes well beyond what has been attainable before. For instance, there was a very recent paper by Hoogland et al. (2015, Curr Biol.) introducing a method that is closely related to the present one (camera-based, measuring the paw positions as well as the whole body angle) and other measurement systems applied to locomotion exist in abundance, typically published alongside main scientific findings (for example, Kiehn 2013). To be clear, I think the authors have made a very careful and high-quality job, but I am doubtful that the advances provided here represent a major leap rather than a minor incremental change to the state-of-the art.

The paper promises to demonstrate a tool that can be used to monitor whole body coordination and 3D limb kinematics. But I find that the analysis of the whole body coordination boils down to computing the angle of the mid-body axis by utilizing the recorded position of the nose and tail, which is not particularly useful from a neuroscience perspective, as the angle is determined by the interplay of a very large number of muscles that could potentially change in different ways in a large number of deficits. In addition, data on the whole body movement is already provided in the solution of [28]. I also find that the stated monitoring of the 3D limb kinematics boils down to a monitoring of the 3D spatial position of the paw. But each position could theoretically be accomplished by many different configurations in the muscles of the limb, and the latter is the type of information that one would need to obtain a useful tool. Figure 3 seem at first glance to indicate that there is indeed some information on the limb joints – however, when I read the paper and the Methods, there are no indications that the joints are actually measured. Hence, I assume that the joint angles are computed out of the 3D paw data, which is problematic. First, if these angles are not measured but computed, the authors must explain where the data in their Figure 3 comes from. For the paper as a whole, this is a major problem since the main applicability of this setup would be if the user could actually track joint angles. Relying on computational models to deduce the 3D limb kinematics is very different from actually measuring it, and reduces the value of this method.

There are also some statements made about the analysis of pcd mice, claiming that this analysis supports that the cerebellum provides an internal model. But there is no evidence here that the cerebellum provides an internal model of any kind. What is seen here is only that block of the normal information processing pathway of various motor systems perturbs the capability of the central nervous system to rely on internal models. On the other hand, all biological motor control is feed-forward in nature, as it needs to deal with long delays, and feed-forward control can be solved only by using internal models. So the only thing learned from this study is that the normal function of the central nervous system in locomotion is dependent on a normally operating cerebellum. The internal models could be generated elsewhere, with the perturbed Purkinje cells destroying the capacity of the system as a whole by removing an otherwise permissive signal to any other system hypothetically responsible for generating/providing the internal model.

*Reviewer #2*:

The authors developed a very nice method to quantify motor performance during free locomotion in mice. They show that in cerebellar ataxia coordination rather than individual movements are affected. This is an elegant and thorough study, but a number of points require attention:

1) The authors used Purkinje cell degeneration (*pcd*) mice to test for cerebellar ataxia. *Pcd* mice start to lose their Purkinje cells during early adulthood – however, later on, other defects appear e.g. the loss of thalamic neurons after P50 and subsequently more widespread neurodegeneration (see Fernandez-Gonzalez, Science 2002 and references cited therein). The use of *pcd* mice at ages varying between p41 and p154, therefore, is expected to result in heterogeneous stages of neurodegeneration. This is not addressed experimentally. Therefore, the authors have to present their data separately for different age groups – especially for mice younger than p50 (no extra-cerebellar neurodegeneration expected) and older (including thalamic and maybe even more neurodegeneration).

2) Dependence of gait parameters of individual limbs on weight and speed.

The authors provide an elegant analysis of the correlations between weight, speed and specific parameters of single-limb movements. As expected, many (if not all) parameters depend on speed. The effect of weight, however, is more prominent in some parameters than in others (e.g. compare Figure 2). The authors conclude that, given body weight and speed, they can make reasonably accurate predictions for single-limb movement parameters. However, the authors do not directly disentangle the impact of body weight vs. speed for each parameter. In other words, how much of the variation is explained by either one? Were there also gender- and/or age-related differences, or were these fully explained by variations in body weight (see Figure 2–figure supplement 2)?

3) Impact of speed for analyzing individual limb gait parameters in *pcd* mice.

The authors show in Figure 3 that *pcd* mice make smaller, slower steps than wildtype littermates. They argue that, despite the actual values being different from controls, individual limb parameters are largely in line with those from control mice at similar speed and with similar weight. This is an important finding, but it would be informative to show the impact of body weight alone. In other words: it seems that the (on-axis) trajectory of single limbs is not really different between *pcd* and control mice, but what about the timing?

In addition, the off-axis movements do seem to be affected, but it is not fully clear to me to what extent corrections for body size and speed have been applied here. In the Discussion, I feel that a better balance between the preserved on-axis parameters and the affected off-axis parameters in the discussion would be warranted.

4) I feel that the remark "the failure of existing systems to quantitatively capture gait ataxia" (in the subsection “Interlimb and whole-body coordination are specifically impaired in *pcd*”) is a bit too strong and does not acknowledge the merits of previous quantifications (e.g. Erasmus Ladder and transparent disk treadmill, which both can also quantify interlimb coordination). Although the authors show that the trajectory of individual limbs is not grossly affected in *pcd* mice, the speed is (Figure 3). And the speed is in turn related to the velocity of single limbs (Figure 2). Thus, without neglecting the importance of the current study, stating that previous studies were not able to quantitate features of gait ataxia or interlimb coordination is in my opinion too strong. For possibilities of transparent disk treadmill to study interlimb coordination, please refer to [28], Current Biol). Here, the level of ataxia in tottering is not only quantified (including interlimb coordination), but also related to the level of complex spike synchrony. Please discuss this neurobiological mechanism upfront as well as the advantages and disadvantages of the technical possibilities of this disk treadmill system with respect to the current locomouse technology.

5) The authors show (in Figure 5) that tail lateral movements of control mice have a sinusoidal appearance, related to the stride phase. This movement is largely exaggerated in *pcd* mice. Whereas the lateral movements in control mice are coupled to the stride phase, in the *pcd* mice there seems to be more of a coupling to the time. The tail movements could be either a compensatory mechanism to counteract imbalance due to improper limb control or they could be affected themselves, contributing to further imbalance. Currently, the variations in tail movements are not really taken into account. Could the authors provide a more solid analysis showing whether tail movements are compensatory or affected by themselves?

6) In the Discussion, the authors mention that previous studies lack the fine-grained detail used in this work and therefore focused on "markers for global motor dysfunction [but] they lack specificity" (subsection “A quantitative framework for locomotor coordination”). Now that the authors have developed such a beautiful system: what is the effect of cerebellar ataxia? How does this separate from other motor dysfunctions (e.g. muscular dystrophy)?

7) In Figure 4, the authors showed the step-cycle for four limb movements by indicating the stance onset for each paw (Figure 4) and the support pattern (Figure 4). However, I find it still difficult to understand the phase relationship between limbs with regard to swing and stance phase. For example, to what extent does the onset of the swing phase of the forelimb occur during the swing or stance phase of the ipsilateral hindlimb? In order to have a better interpretation of inter-limb coordination, it might be worthwhile to add an additional figure with swing and stance phase indicated for each individual limb.

Reviewer #3:

The authors of this study have developed an automated kinematic analysis of locomotion in mice. They use it for locomotor analysis in freely moving wild-type mice. A detailed description of locomotor parameters allowed them to create a mathematical model to predict most of the parameters knowing just the walking speed and the body size of the animal. They extend this analysis to a mouse model for Purkinje cell degeneration (*pcd*). They conclude that the traditional locomotor parameters (stride length, cadence, intra-limb coordination) and forward trajectories are not different in wild-type mice and *pcd* mice. In contrast they conclude that the whole-body coordination is specifically impaired in *pcd* mice. Furthermore, a specific description of nose and tail oscillations in *pcd* mice were modeled as passive consequences of the forward motion of the hindlimbs. The main conclusions from the study are: a) only by studying whole-body coordination will it be possible to reveal the ataxic locomotor phenotype, and b) the observed locomotor changes are consistent with the hypothesis that the cerebellum provides a forward model for motor control. Although I find the study of great interest there are a number of issues that are problematic for the interpretation of the data. The authors will have to address these issues to avoid making incorrect statements about the findings and the method used.

1) The authors make a big deal out of using machine learning as an 'objective' way of measuring locomotor parameters. But there seems to be a lot of noise in the detection. This is seen in Figure 1. Where is the level set for step onset and offset? Figure 2 is an even stronger case. At very low speed (<0.4) the stride length varies with a factor of 10 from 1 cm to 10 cm. 10 cm long stride lengths are not common at these speeds in mice. It must be a detection mistake. Also there are many values with very low stride lengths spread over all speeds. One would like to know how much irrelevant noise that the automated procedure introduces for example by showing traces from the low speed locomotion with 1 cm and 10 cm stride lengths. Everything in Figure 2 is known from many previous studies of rodent locomotion – so there is nothing new about it except that the authors have analyzed a large amount of steps.

2) The authors make a point out of not seeing any intralimb differences between wild-type and *pcd* mice. Looking at Figure 3 this referee is not convinced, and it seems that this statement that is likely to be wrong. There seems to be clear differences in almost all parameters. Statistical tests will be needed to support the claims.

3) A main conclusion is that the interlimb coordination changes from wild-type to *pcd* mice. More specifically from a trot to a walk pattern. Walking and trotting patterns have been described for tetrapods to be distinct gaits by many different studies. They are visible at different speeds of locomotion. In your experiments wild-type mice exhibited trot as unique gait used at all speeds, even at the lowest ones when the speed is below 0.2 m/s and the support limbs are three or four (Figure 4). In contrast the *pcd* mutant mice exhibit only walk, reaching a maximum speed of 0.25 m/s. What is odd about this difference is that walk is clearly found in wild-type mice by all others than these investigators. I believe that it is present even in the present study but that it has been missed (how can you have trot with 3 or 4 limb on the ground?). Since the *pcd* mice locomote slower than the wild-type mice the change in coordination could simply be caused by a change in gait patterns that is speed dependent. It is known for many animal models (mouse as well) that the inter-limb and whole body coordination change when the animal uses different gaits. At the mechanical level different models have been used to explain the different body coordination of walking and trotting gaits (Mechanical work in terrestrial locomotion: two basic mechanisms for minimizing energy expenditure, Cavagna et al., 1977, American Journal of Physiology). Thus, most of the differences you describe between wild-type and mutant mice could be due just to the comparison of mice using two different gaits that clearly rely on different inter-limb and whole body coordination (e.g. Figures 4 and 6). Indeed you have clear indications that walk is also expressed in wild-type mice at low speed (Figure 4). The data should be reanalyzed to address this issue. I am convinced that if the data are reanalysed, walk will show up in wild-type mice as well. This will dramatically change the conclusion of the story.

4) The discussion about the superiority of the present method to others needs to be toned down. The general statement that the automated analysis described in the study provides a robust way for locomotor analysis avoiding "false positive findings that permeate the mouse locomotor literature" is not true and likely to offend the locomotor field at large. I strongly recommend that the authors consider gaits (walk, trot, gallop and bound) in their analysis. Since you have done an incredible work using size-matched controls, it will be helpful if the difference between wild-type and *pcd* mutant mice account just for the degeneration of the Purkinje cells and not for errors due to a comparison between mice using two different gaits.

5) To my knowledge *pcd* mice showed not only a degeneration of the Purkinje cells but also the slower degeneration of the photoreceptor cells of the retina and mitral cells of the olfactory bulb. Since you have used animals between 41 and 154 days old, could it be that a shorter range of speed is the result of a disorganization of the visual system and not only due to an impairment of the neural control of the locomotion? They walk slower because they are half blind?

[Editors’ note: what now follows is the decision letter after the authors submitted for further consideration.]

Thank you for submitting your work entitled "A quantitative framework for whole-body coordination reveals specific deficits in freely walking ataxic mice" for peer review at *eLife*. Your submission has been favorably evaluated by Eve Marder (Senior Editor), a Reviewing Editor, and three reviewers.

The reviewers have discussed the reviews with one another and the Reviewing Editor has drafted this decision to help you prepare a revised submission.

The reviewers all appreciated the fact that the novel LocoMouse tool permits automated analysis with high resolution of locomotion of freely walking mice, and they judged that this method has the potential to be of value to the field. They provide a number of comments for strengthening the manuscript, which are included below. These comments center on the being more explicit about the limits of the work, its interpretation (especially regarding the mutants), and comparison to other studies, providing more information about certain analyses, and addressing some points for clarity of presentation and communication. Specifically:

1) Toning down the claims about the superiority and power of the new method, perhaps providing a more explicit comparison to commercial packages.

2) Toning down the claims about providing direct evidence for forward models, perhaps simply by shifting these ideas to the Discussion.

3) Limiting the claims about changes and the lack of changes in the mutant, perhaps by focusing on the idea that one kinematic measure was preserved while others (interlimb and intralimb) were not.

4) Providing a better description of the construction of the passive model, and (if feasible) providing a physical model (or explaining why this was not done).

5) Addressing the point either experimentally or with clarification of the (apparently) very low N (=3) for the *pcd* mice.

6) Improving presentation of the results/figures/sequence of calling figures, as indicated in the reviews.

Reviewer #1:

1) The authors emphasize in several places that 'traditional' measures of kinematics are unimpaired in mutants. I found this to be overly-simplistic, at least based on the results presented here. As I understand it, the main consistency is in the x-axis forward paw motion, while there are changes in vertical and mediolateral paw motion and in joint angles. Changes in vertical paw trajectory and joint angles are hardly 'non-traditional', 'complex', or even '3D' and all measures are definitely 'individual-limb'. This point is made several times in the text (see points below) and seems unwarranted. I'm not saying the results aren't interesting, nor do I think that these conclusions are necessary for the impact of the paper – I just don't think that this conclusion as stated is supported.

2) On a similar point, it's not clear to me that any of their measures are really looking at within-limb coordination, at least as this is usually considered. Most of the measures relate to paw movement, whereas traditionally issues of coordination (e.g. in studies examining the role of the cerebellum, as cited in the paper) consider interactions between joints. Yet these analyses really aren't done here. They show that joint angles are apparently different in the animals, but these are for single joints as opposed to joint angle relationships (and what those angles are is unclear, see below). I don't think that the results shown here rule out a possible deficit in traditional, within-limb coordination. I think this is mainly an issue of presentation, however, rather than one of substance. As stated above, I don't think the impact of the paper relies on this conclusion and it could be made more precise without loss of significance.

3) I potentially like the analysis of nose and tail passive movements very much, but I did have a few issues with it. First, although the results are consistent with a role of the cerebellum in predictive control based on forward models, they're not really a direct demonstration thereof. A more direct demonstration might involve examining adaptations to perturbations, e.g. adding a weight to the tail and seeing that initially control animals look like mutants but then learn to predict and compensate for this, whereas mutants never do so. Also, could the results simply be interpreted as one of a limited capacity for motor control? e.g. mutants have limited capacity for producing locomotion and expend most of it on x-axis movement of the paw since it's critical for task performance, whereas other aspects of behavior are allowed to vary? I appreciate the authors are careful on this point to say that the results are 'consistent' with this idea, but alternative explanations might be considered more directly.

4) For the nose/tail passive analysis, I originally thought that this was an actual physical model, rather than the geometric model used here. It seems clear that a physical model is the better way to do this. Why wasn't it done? The geometric model, with its assumptions of lags taken from data which is actually observed, seems more indirect and therefore less compelling. Finally, as I understand it (and I might have missed it), this analysis was only done for the mutants. It would be very good to have done the same analysis for the control animals and show that the tail/nose movements are inconsistent with passive movement, e.g. it could have been that the movement of the pelvic girdle was different in controls, leading to less tail movement (even though the hindlimbs are in alternation for both controls and mutants).

Reviewer #2:

The paper by Machado et al. examines gait patterns in wildtype and Purkinje cell degeneration (*pcd*) mice using a novel system for tracking locomotor kinematics in freely walking mice. The authors conclude that the individual limb kinematics of *pcd* mice are normal when the weight and the walking speed of the mice are taken into account. In contrast, *pcd* mice have impaired coordination of movements across joints, limbs and body. These results are interpreted as evidence that the cerebellum provides a forward model. This is a strong paper. I commend the authors for wrestling with a complex dataset and extracting from it a number of thought-provoking findings. I personally found many of the results exciting, but I must admit that I'm not an expert in the field of locomotion or ataxia. It was not always clear if a particular result in the paper was a major advance or just confirmation of something we already knew. For this reason, I think that it's important for the authors to clarify and emphasize the significance of their results throughout the text. I've given two examples below:

1) It is remarkable that in wildtype mice, one can account for upwards of 80% of the variance in kinematic parameters like stride length, swing velocity and cadence, simply by taking into account the walking speed and the weight of the mouse. Is this a new approach? In their rebuttal, the authors mention that the [11] paper found that all the gait differences between lurcher and control mice disappeared when they corrected for walking speed. Did Cendelin et al. make the correction using a similar modeling approach?

2) One of the main findings is that individual limb movements of *pcd* mice are relatively spared (when weight and walking speed are taken into account), whereas interlimb coordination is impaired. It is not clear how much of this we already knew. For example, in the subsection “Front-hind Interlimb coordination is specifically impaired in *pcd*”, the authors mention that previous studies in ataxic mice have failed to detect any impairments in the kinematic parameters of individual limbs. Did those studies not examine interlimb coordination as well? If the authors are the first to show that mice with cerebellar deficits have a specific impairment in locomotor coordination but not in movement of individual limbs, this is a big deal.

Additional recommendations to help make an already strong paper even stronger:

3) Drop the emphasis on forward models. The authors argue that the deficits they've uncovered point to a lack of a forward model in *pcd* mice. Although I found some of the evidence presented intriguing (for example the analysis of tail movements), it is far from conclusive. Furthermore, it is difficult to reconcile the normal individual limb kinematics with a faulty forward model – individual limbs are controlled by multiple agonist/antagonist muscles whose coordination requires well-calibrated forward models. I think the authors can still interpret their results in the context of forward/inverse models, but I strongly recommend that they save this interpretation for the Discussion, tone it down a bit and address the limitations. The paper would be stronger if the second paragraph of the Introduction is rewritten to highlight the main questions to be addressed in the paper, instead of offering to resolve controversies about forward vs inverse cerebellar models.

4) Analysis of variability: All the analyses are based on average data. For example, Figure 3 shows that there is no difference in the individual limb kinematics of *pcd* mice and sized-matched controls. But this is done only for the average kinematic values. Is it not possible that *pcd* mice might show increased trial-to-trial variability in their movements relative to sized-matched controls? How would such a finding change the authors' conclusions?

Reviewer #3:

The authors developed a machine learning algorithm to track the individual limbs as well as the nose and tail of mice in 3D during natural, voluntary locomotion. They provide a very detailed analysis of locomotion parameters in control and ataxic Purkinje cell degeneration (*pcd*) mice, revealing that individual limb kinematics in *pcd* mice are not different from control mice when accounting for the mouse's weight and speed, but that *pcd* mice present multi-joint and inter-limb-coordination deficits. The study has been carefully conducted and provides new and useful tools for measuring and analyzing specific features of locomotion and ataxia with unprecedented accuracy.

1) The construction of the "passive model" (Figure 5) is unclear. For example, does this model take into account the mass of the tail? Please describe in more detail how the passive prediction is calculated.

2) The authors contrast the forward model with an inverse kinematic model in the Introduction. What do they think the inverse kinematic model would predict and how does this contrast with their findings?

[Editors' note: further revisions were requested prior to acceptance, as described below.]

Thank you for resubmitting your work entitled "A quantitative framework for whole-body coordination reveals specific deficits in freely walking ataxic mice" for further consideration at *eLife*. Your revised article has been favorably evaluated by Eve Marder (Senior Editor), a Reviewing Editor, and three reviewers. The manuscript has been greatly improved and the revision was met with enthusiasm. There are only a few remaining issues that need to be addressed before acceptance, as outlined below:

In particular, the manual joint-angle analysis shown in Figure 3 seems limited by the single animal in each condition, and discussion among reviewers and editors led to the consensus that it could simply be dropped without major impact on the manuscript. We therefore recommend cutting this one measurement and illustration. Alternatively, if there is some reason why a single animal in each category is of value, please justify the experiment clearly. This point is explained further below.

In the subsection “Individual limb parameters”, the description makes it seem like one *pcd* animal and one control were used for this analysis. If so, this seems very questionable and underpowered. I'd be willing to bet that, with enough n, it would be possible to find a significant difference between two different weight matched controls. I'm not sure how the stats behave in this situation, but it's not clear to me how mixed models would be able to partition variance due to random effects (individual subjects) from the fixed effects (mutation) if there were only one animal in each group. It seems that for other comparisons with weight matched controls much larger N's are used, whereas this analysis is limited because of the need to hand mark frames. It might be easiest to simply drop this joint angle analysis – I don't think its omission would change the impact of the paper. Also, shouldn't it be ‘knee, heel, toe’? And here it's stated that the angle of the foot and arm relative to the ground are quantified – I might have missed it, but I wasn't sure where that was reported in the Results.

---

## [Author Response]

[Editors’ note: the author responses to the first round of peer review follow.]

We are confident that we have addressed the reviewers’ concerns, without further experiments or substantial re-analysis of the data. Moreover, we are grateful for constructive criticism and were happy to revise the manuscript to address the concerns and questions raised by all three reviewers. However, while many of the specific comments were helpful, we have some issues with nearly every point raised in the summary paragraph that provides the rationale for the decision on our manuscript.

Specifically, we note that our reviewers heavily refer (both explicitly and implicitly) to two papers that came out in press in Current Biology *after* we first submitted our paper to *eLife*. While any rejection is difficult, we feel that our case warrants an appeal. Please find a detailed response to the concerns raised during review below.

All the reviewers find that the study is very interesting and that the method presented for monitoring locomotion in relation to the cerebellar motor function and other motor functions is potentially strong. There are, however, concerns whether the automation procedure is making correct assessments of the locomotor parameters […]

We address this point in detail below in response to Reviewer #3.

[…] that gaits are not taken into accounts in the description,

We address the issue of categorizing interlimb coordination with gaits in detail below in response to Reviewer #3.

[…] that there is a lack of statistical evaluation of intra-limb coordination,

Figures 2 and 3 and their supplements are entirely devoted to the statistical evaluation of intra-limb coordination. We address this very important point below in response to the reviewers’ specific comments.

[…] and that the differences in full body movements may not account completely for the phenotype.

We address these very important points in more detail below. To be clear, we agree that *there are clearly differences in individual limb movements in* pcd, as is shown in Figure 3. However, we also show that many of these differences (specifically, all of the ones related to forward paw motion) are fully accounted for by the statistical models derived in Figure 2 that predict gait parameters based solely on walking speed and body size. The thick lines in Figure 3 are not fits to the data, they are the values of gait parameters that are *predicted* based on the models generated in Figure 2, for this new set of mice, walking at these speeds. The fact that they match the data for both *pcd* and littermates, demonstrates that *the differences in these parameters in* pcd *(while real!) do not depend on nor account for the ataxic phenotype*. In contrast, the off-axis difference in *pcd* cannot be accounted for by differences in walking speed or body size, and so do reflect ataxia.

To meet these concerns would require substantial reanalysis of the data and reconsidering the main conclusions of the study.

We don’t think it does. We had already performed nearly all of the requested analyses – most we had not included simply out of concern for length. We have now also performed additional analyses to address specific concerns, many of which we include here to show that the main conclusions of the study are not changed.

Even with a new analysis there is a feeling that the results presented will mainly be a technical advance rather than a conceptual advance for cerebellar motor function or the locomotor field at large.

We respectfully, yet strongly, disagree. We believe that our mathematical modeling approach, our quantitative treatment of interlimb phasing, the passive geometric modeling of nose and tail oscillations (which perfectly match those of *pcd*), and our correlation analysis of coordination across the body, all provide an entirely new quantitative and conceptual framework for understanding mouse locomotor coordination. This framework allowed us not just to capture but also isolate specific deficits in multijoint, interlimb, and whole-body coordination in ataxia. We started with a messy, entirely unconstrained dataset, but through the application of quantitative principles, it revealed a very specific deficit in *pcd* mice – an absence of forward models that allow the mouse to control the movements of individual parts of the body (joints, limbs, head, and tail) through the use of predictions about the movements other parts of the body (joints, limbs). The major effort here, and the major advance, was in using quantitative analysis to establish a conceptual framework that yields meaningful insights into biological data.

Reasons for this are that there already are cerebellar specific mutants that have been analyzed in similar ways […]

We are of course, happy to update the manuscript and include this recent paper (Hoogland et al.), as requested by Reviewer #2. However, we also point out that our paper – in its goals, approach, and conclusions – is quite different from that of Hoogland et al., not least because we are concerned with analyzing locomotor coordination in freely walking mice. Moreover, beyond being against *eLife’s* policy of assessing novelty, since the papers were near-simultaneous, the analyses in the two papers are so different that we’re not sure how anyone who has read both papers can argue that they are similar enough to interfere with the conclusions of our paper.

[…] and that the locomotor data in wild-type are confirmative and lacking gait parameters.

We disagree and outline our arguments in detail below in response to Reviewer 3.

*For these reasons the manuscript as presently configured is not suitable as a regular paper in* eLife*. The details of the concerns and suggestions for improvements are detailed in the attached review reports*.

eLife *has "Tools and Resources" that present methods papers. If you feel that you will be able to reanalyze the data as outlined and make a case that this method is better and more usable than other methods already published it might be an option for you to consider submitting your work in this form to* eLife.

Thank you. We appreciate that this paper could be seen as either a Tools or a Research Article. We hope, given the present clarification of our rationale and the fact that we have now addressed the reviewers’ specific concerns, through clarification of misunderstandings, additional analyses, and revisions to the manuscript (outlined point-by-point below), that you will be willing to provide us guidance on how to proceed with possible publication in *eLife*, in whichever format you find appropriate.

Reviewer #1:

*The paper of Machado et al. introduces a camera-based motion capture system for the analysis of walking movements in the mouse. The heart of the system is machine learning operating on resampled images of high speed cameras for mice walking in a glass corridor. To illustrate its applicability, the paper compares the walking movements of wildtype and Purkinje cell degeneration (*pcd*) mice*.

*In general, for a paper introducing a novel analysis tool, I think it must be assessed whether the method provides measurements of highly useful parameters, with high explanatory value, that goes well beyond what has been attainable before. For instance, there was a very recent paper by Hoogland et al. (2015, Curr Biol.) introducing a method that is closely related to the present one (camera-based, measuring the paw positions as well as the whole body angle) and other measurement systems applied to locomotion exist in abundance, typically published alongside main scientific findings (for example,*
*Kiehn 2013**). To be clear, I think the authors have made a very careful and high-quality job, but I am doubtful that the advances provided here represent a major leap rather than a minor incremental change to the state-of-the art*.

We point out that this paragraph makes the implicit assumption that the tracking system we introduce is meant to be the main advance in the paper. For us the tracking system was a tool that we needed (and that did not and does not to our knowledge exist) to establish a quantitative framework for whole-body coordination *in freely walking mice*. We provide the full details of our system because we want it to be of use to other labs interested in the neural control of coordinated movement. But to be very clear, we think the major advance here is a new conceptual framework for mouse locomotor coordination. It was made possible by the increased resolution and throughput of the new tracking system that we describe, but more importantly, the isolation of specific features of ataxia in freely walking mice provides novel insights into mouse locomotor coordination that challenge the existing state-of-the art.

At the same time, we admit that we had previously failed to express that we recognize the value of existing systems for measuring locomotion and we know that these have been used successfully to analyze contributions of the spinal cord, in particular, to the generation of locomotor patterns. We have edited the text of the manuscript to reflect this appreciation more explicitly (in the subsection “LocoMouse”). However, critically, these approaches have not been nearly as successful in providing insights into cerebellar contributions to locomotion. The point we are trying to make here is that these failures have been for three main reasons: 1) a failure to distinguish between specific features of ataxia vs. secondary consequences of slower walking speeds and smaller body sizes, 2) a lack of detailed, 3D kinematic data, and 3) a lack of emphasis on coordination across joints, limbs, and body, which are so particularly sensitive to cerebellar damage and which are not the main focus of most off-the-shelf locomotor assays. We hope that our revisions make our position more clear on these points.

*The paper promises to demonstrate a tool that can be used to monitor whole body coordination and 3D limb kinematics. But I find that the analysis of the whole body coordination boils down to computing the angle of the mid-body axis by utilizing the recorded position of the nose and tail, which is not particularly useful from a neuroscience perspective, as the angle is determined by the interplay of a very large number of muscles that could potentially change in different ways in a large number of deficits*.

First, we use whole-body coordination here not just to mean measuring nose and tail movements, but also *the spatiotemporal relationships across various body elements*, as in Figure 6. Here, whole-body coordination is demonstrated by the ribbon plots and correlation matrices in a way that is meant to highlight that the spatiotemporal relationships of movements across the body are impaired in *pcd* (in contrast to the forward paw motion, which is intact, Figure 3). We have changed the headings in the Results to be more consistent with our true meaning.

Second, we are confused by the reviewer’s claim that our “analysis of whole-body coordination boils down to computing the angle of the mid-body axis”, as this is not something we report. Perhaps they have confused our analysis with that of Hoogland et al., to which this comment more readily applies? Or perhaps they were misled by looking at Figure 5 (now 5C)? To clarify, Figure 5*(C) does not show how we measured the tail, but rather, shows how we derived the equations predicting what passive oscillations of the tail would look like if they resulted solely from the orthogonal coupling between the forward motion of the hindlimbs and side-to-side tail movement*. Our measurements of tail movements were, as described in the methods and figure legend, 3D position values for 15 individual tail segments. We completely agree that the movements of the tail could potentially change in different ways in a large number of deficits, *which is precisely why the demonstration that the tail oscillates as a passive consequence of hind limb motion in* pcd *is so important*. This wasn’t at all trivial and together with the off-axis/multijoint impairments of individual limbs, points to a specific lack of ability to predict the consequences of movements of other parts of the body.

Finally, we respectfully disagree with the reviewers’ apparent position that quantitative analysis of kinematics is not useful for neuroscience. While ultimately, of course, movements are effected by muscles, kinematic analysis has been fundamental to the understanding of the neural control of movement, and *kinematic representations have been found in many motor areas, in particular in Purkinje cells*, where they have been dissociated from muscle activity/dynamics (46). Moreover, our analyses of 3D paw trajectories and joint angles actually highlight the fact that yes, absolutely, there are different ways to achieve the same movement, and our finding that *pcd* mice achieve the same forward movements of the paws with different off-axis movements and different joint angles, we would argue, is in itself an argument for the importance of the kind of detailed kinematic analysis that we provide.

In addition, data on the whole body movement is already provided in the solution of [28].

We now cite the now-published Hoogland et al. paper from the De Zeeuw group in our manuscript. However we point out a very meaningful difference between our study and theirs, which is that ours is in *freely walking mice*. There are likely to be critical differences in whole-body coordination in these different setups. Moreover, again, we do not see the only advance in our paper as being the measurement of movement per se, but rather emphasize the quantitative and conceptual framework we use to analyze it.

*I also find that the stated monitoring of the 3D limb kinematics boils down to a monitoring of the 3D spatial position of the paw. But each position could theoretically be accomplished by many different configurations in the muscles of the limb, and the latter is the type of information that one would need to obtain a useful tool*.

We take the reviewer’s point to be more clear in the text about when we are referring to limb vs. paw kinematics, and have changed several sentences accordingly. However, we also reiterate our belief, stated above, that kinematic analysis has been generally useful for motor control and that the specificity of the tail and 3D paw findings in *pcd* are themselves strong arguments that detailed kinematic analysis can provide important insights.

Figure 3
*seem at first glance to indicate that there is indeed some information on the limb joints – however, when I read the paper and the Methods, there are no indications that the joints are actually measured. Hence, I assume that the joint angles are computed out of the 3D paw data, which is problematic. First, if these angles are not measured but computed, the authors must explain where the data in their*
Figure 3
*comes from. For the paper as a whole, this is a major problem since the main applicability of this setup would be if the user could actually track joint angles. Relying on computational models to deduce the 3D limb kinematics is very different from actually measuring it, and reduces the value of this method*.

We apologize that we were not clear about this. The *joint angles were absolutely measured, not computed*. We agree that this would have been problematic. We have added the joint angles to the list of measurements in the Methods and have rewritten the relevant paragraph in the subsection “Single-limb deficits in *pcd* are restricted to off-axis paw trajectories and joint angles”. The joint angles were computed manually by clicking on movie frames.

While more labor intensive than our other analyses, the only way we could have made the joint tracking automatic would have been to use markers, and we think the advantages of a markerless system outweigh any potential concerns about reduced throughput – and in fact the throughput here is similar to other non-automated systems (eg Motorater, Zorner et al.).

*There are also some statements made about the analysis of* pcd *mice, claiming that this analysis supports that the cerebellum provides an internal model. But there is no evidence here that the cerebellum provides an internal model of any kind. What is seen here is only that block of the normal information processing pathway of various motor systems perturbs the capability of the central nervous system to rely on internal models. On the other hand, all biological motor control is feed-forward in nature, as it needs to deal with long delays, and feed-forward control can be solved only by using internal models. So the only thing learned from this study is that the normal function of the central nervous system in locomotion is dependent on a normally operating cerebellum. The internal models could be generated elsewhere, with the perturbed Purkinje cells destroying the capacity of the system as a whole by removing an otherwise permissive signal to any other system hypothetically responsible for generating/providing the internal model*.

a) We respectfully disagree that all biological motor control is feedforward – feedback is often used to adjust ongoing movements, albeit with a delay.

b) Although we personally agree with the reviewer about the importance of internal models for motor control*, there is significant controversy on this point within the cerebellar field, in terms of whether the cerebellum provides inverse or forward models, or both, or neither*. See, for example, the second paragraph of our Introduction, and the references within it.

c) Our data point *specifically* to a lack of a *forward* (but not inverse) model in ataxic *pcd* mice. Why? Because i) Forward paw kinematics are normal, which would not be the case if there was a failure in an inverse model. ii) Previous studies have argued that a lack of a forward model leads to deficits in multijoint coordination in ataxic patients (e.g. [4]), very like what we see here in Figure 3) The passive oscillation of the tail in *pcd* reveals the lack of a mechanism that normally predicts and actively cancels this oscillation in wildtype mice. We have edited this portion of the discussion to be more specific on these points.

d) While it is true that cerebellar output could be permissive for the function of a forward model rather than generating/providing that model per se, the hypothesis that it actually generates it has been influential in the field. We had tried to be careful with our wording to avoid going beyond our data on this point (e.g. from the subsection “Cerebellar contributions to coordinated movement”: “The changes in multijoint and interlimb coordination, and the apparent passive oscillations of the nose and tail in *pcd* can be interpreted as a failure to predict the consequences of movements of other parts of the body, an idea that is consistent with forward model theories of cerebellar function (4; 18; 29; 31; 57).” We have edited several other statements to avoid words like “generating” as the reviewer suggests. Still we emphasize that the fact that all of the deficits we observe can be interpreted as the failure of a forward model points to this being an important aspect of cerebellar function, permissive or not.

e) We think the statement, “the only thing learned from this study is that the normal function of the central nervous system in locomotion is dependent on a normally operating cerebellum” is unfair. The point of this paper is to overturn that notion and show that actually, *locomotion is affected in very specific ways in* pcd. The preservation of forward paw kinematics was completely unexpected based on the existing literature and it changes the way we think about ataxia. It also reveals that while Purkinje cell activity often correlates with movement kinematics, this may not be its essential function – at the very least, other parts of the brain can compensate. However other parts of the brain do not appear to compensate for the loss of a forward model.

Reviewer #2:

*The authors developed a very nice method to quantify motor performance during free locomotion in mice. They show that in cerebellar ataxia coordination rather than individual movements are affected. This is an elegant and thorough study, but a number of points require attention*:

Thank you and we address all of these points below.

*1) The authors used Purkinje cell degeneration (*pcd*) mice to test for cerebellar ataxia.* Pcd *mice start to lose their Purkinje cells during early adulthood – however, later on, other defects appear e.g. the loss of thalamic neurons after P50 and subsequently more widespread neurodegeneration (see Fernandez-Gonzalez, Science 2002 and references cited therein). The use of* pcd *mice at ages varying between p41 and p154, therefore, is expected to result in heterogeneous stages of neurodegeneration. This is not addressed experimentally. Therefore, the authors have to present their data separately for different age groups – especially for mice younger than p50 (no extra-cerebellar neurodegeneration expected) and older (including thalamic and maybe even more neurodegeneration)*.

Yes, we agree that it is an important consideration and we have done this. We have added some graphs as a figure supplement (Figure 5—figure supplement 2) that show that the main conclusions of our study hold in the youngest *pcd* mice (<p50).

2) Dependence of gait parameters of individual limbs on weight and speed.

*The authors provide an elegant analysis of the correlations between weight, speed and specific parameters of single-limb movements. As expected, many (if not all) parameters depend on speed. The effect of weight, however, is more prominent in some parameters than in others (e.g. compare*
Figure 2*). The authors conclude that, given body weight and speed, they can make reasonably accurate predictions for single-limb movement parameters. However, the authors do not directly disentangle the impact of body weight vs. speed for each parameter. In other words, how much of the variation is explained by either one? Were there also gender- and/or age-related differences, or were these fully explained by variations in body weight (see Figure 2–figure supplement 2)*?

We apologize that we were not more clear on this point, because we did in fact directly quantify the relative contributions of body weight, speed, age and gender. These were reported in Figure 2–figure supplement 2, but clearly we should have been more explicit. To address this, we have edited the treatment of this issue in the subsection “Gait parameters for individual limbs vary consistently with walking speed and body size” and rewritten the Legend for the figure supplement to explain in less technical terms what we had done. We have also added specific demonstrations of the failure of the additional factors of gender or age to improve the predictions above and beyond body weight and walking speed alone, both to the tables in the supplement and in the text.

*3) Impact of speed for analyzing individual limb gait parameters in* pcd *mice*.

*The authors show in*
Figure 3
*that* pcd *mice make smaller, slower steps than wildtype littermates. They argue that, despite the actual values being different from controls, individual limb parameters are largely in line with those from control mice at similar speed and with similar weight. This is an important finding, but it would be informative to show the impact of body weight alone. In other words: it seems that the (on-axis) trajectory of single limbs is not really different between* pcd *and control mice, but what about the timing*?

First, thank you for pointing out the importance of this finding, which we agree is a major contribution of the paper. In terms of showing the impact of body weight alone, we point out that in Figure 3, the differences in the model predictions (thick lines) for the two sets of mice are due entirely to body weight differences, since speed and weight were the only two factors included in the predictions, and the data are plotted as a function of speed. We don’t understand the last sentence, however, about the timing – if you mean interlimb timing, we address that point in Figure 4 (and below in response to Reviewer 3). Or do you mean the temporal profile of the trajectories themselves? Those are shown in Figure 3, for *pcd* and size- matched controls.

To emphasize the role of body weight alone, and to further illustrate the fact that changes in body weight and walking speed together can fully account for the changes in forward limb movement in *pcd*, we have added a Figure Supplement to Figure 3 that compares the gait parameters of *pcd*, littermates, and size-matched controls.

In addition, the off-axis movements do seem to be affected; but it is not fully clear to me to what extent corrections for body size and speed have been applied here.

This is an important point, but please note that the last line of the legend for Figure 3 states, “For E-H controls are size-matched control […] see Figure 2—figure supplement 1”, and that at the end of the subsection “Changes in gait parameters for individual limbs are predicted by changes in walking speed and body size” we state: “*pcd* animals are compared with size-matched controls from here on.” Similarly, speed is indicated by the line thicknesses on each trajectory plot and the legend for Figure 3 states “line thickness represents increasing speed”. To be sure to avoid any misunderstanding on this important point, however, we changed the color-coding in Figure 3 to distinguish between littermates and size-matched controls and we have added the words “size-matched” before “controls” at many places in the text and legends. We also now specify that the line thickness legend in the panel of Figure 3 refers also to 3F and 3G.

In the Discussion, I feel that a better balance between the preserved on-axis parameters and the affected off-axis parameters in the discussion would be warranted.

Yes, we agree, and we have revised this part of the Discussion, based on the feedback from this reviewer and Reviewer #3. We also added a new, separate heading to the Results.

*4) I feel that the remark "the failure of existing systems to quantitatively capture gait ataxia" (in the subsection “Interlimb and whole-body coordination are specifically impaired in* pcd*”) is a bit too strong and does not acknowledge the merits of previous quantifications (e.g. Erasmus Ladder and transparent disk treadmill, which both can also quantify interlimb coordination). Although the authors show that the trajectory of individual limbs is not grossly affected in* pcd *mice, the speed is (*Figure 3*). And the speed is in turn related to the velocity of single limbs (*Figure 2*). Thus, without neglecting the importance of the current study, stating that previous studies were not able to quantitate features of gait ataxia or interlimb coordination is in my opinion too strong*.

We have toned this down in several ways. First, we are more explicit about the contributions of the other methods, such as the ones mentioned by the reviewer. Second, we are more explicit about when we are referring specifically to gait patterns of *freely walking* mice, which neither the Erasmus ladder nor the transparent disk treadmill address. Third, and most importantly, we clarify several places in the text to make it more clear that out that our point is not that other studies have not *measured* changes associated with gait ataxia, but rather that: a) those changes do not necessarily relate to ataxia per se, beyond reflecting changes in walking speed alone, which are not specific and b) that here, we not only measure changes associated with ataxia that have not previously been reported (3D trajectories, specific front-hind interlimb coordination deficits, and tail movements), but we also isolate them from the non-specific effects of altered walking speed. Finally, we point out that the specific sentence mentioned by the reviewer, was meant to refer specifically to the reference within it (11), which attempted to quantify ataxia in freely walking lurcher mice with the CatWalk system, and found that all the differences between lurcher and control mice that they observed disappeared when they corrected for walking speed. But we didn’t say this explicitly and we realize that not everyone who reads our paper will be familiar with that work – hence the changes to the text that we have made in response to the reviewers.

*For possibilities of transparent disk treadmill to study interlimb coordination, please refer to*
[28]*, Current Biol). Here, the level of ataxia in tottering is not only quantified (including interlimb ccordination), but also related to the level of complex spike synchrony. Please discuss this neurobiological mechanism upfront as well as the advantages and disadvantages of the technical possibilities of this disk treadmill system with respect to the current locomouse technology*.

We have added the now-published [28] to our revision.

*5) The authors show (in*
Figure 5*) that tail lateral movements of control mice have a sinusoidal appearance, related to the stride phase. This movement is largely exaggerated in* pcd *mice. Whereas the lateral movements in control mice are coupled to the stride phase, in the* pcd *mice there seems to be more of a coupling to the time. The tail movements could be either a compensatory mechanism to counteract imbalance due to improper limb control or they could be affected themselves, contributing to further imbalance. Currently, the variations in tail movements are not really taken into account. Could the authors provide a more solid analysis showing whether tail movements are compensatory or affected by themselves*?

We have edited the text of the results to make it more clear that our geometrical model addresses the reviewer’s point quite nicely. The model, shown by the grey dashed lines in Figure 5 and the thick lines in Figure 5, calculates *predictions of what passive oscillations of the tail would look like if they resulted solely from the orthogonal coupling between the forward motion of the hindlimbs and side-to-side tail movement*. The fact that the grey dashed lines in Figure 5 and the thick lines in Figure 5 match the *pcd* data, reveals that the side-to-side tail oscillation in *pcd* is exactly what is predicted as a passive consequence of forward hindlimb motion. Criticallly, we emphasize *that hindlimb alternation is unaltered in* pcd (Figure 4), and so these changes in tail movement cannot result from changes in hindlimb oscillation. (On a side note, however, the increased oscillations certainly contribute to imbalance, and in fact we think this may be a major reason for the characteristic increased base of support in ataxia that we observe in Figure 3.)

We recognize that we did not adequately explain/emphasize the importance of the passive tail model with our previous submission. To emphasize this important finding, we have reordered the panels in Figure 5, rewritten the description of Figure 5 in the main text to go more slowly through it, and added a movie of the model, to show just how well passive deflections of the tail and nose based solely on perpendicular coupling to forward motion of hindlimbs captures the gestalt of the ataxia in *pcd*.

*6) In the Discussion, the authors mention that previous studies lack the fine-grained detail used in this work and therefore focused on "markers for global motor dysfunction [but] they lack specificity" (subsection “A quantitative framework for locomotor coordination”). Now that the authors have developed such a beautiful system: what is the effect of cerebellar ataxia? How does this separate from other motor dysfunctions (e.g. muscular dystrophy)*?

This is an important point that highlights the potential impact of our approach. Clearly we have isolated specific deficits in coordination across joints, limbs, and body in *pcd*. We very much hope that the current paper will help many groups interested in mouse models of various diseases to identify and isolate specific features of motor function that are affected in their systems. We would speculate, for example, that the sparing of forward paw trajectories in *pcd* would not be expected in the muscular dystrophy example, but interlimb coordination could be spared. We are currently analyzing several other mouse mutants with extra-cerebellar phenotypes and while those studies are in preliminary stages and outside of the scope of the current paper, the results are quite different from those we report here. We further note that *it is not just the tracking system itself that allows for this specificity, it comes especially from the quantitative framework we use to analyze the data*.

*7) In*
Figure 4*, the authors showed the step-cycle for four limb movements by indicating the stance onset for each paw (*Figure 4*) and the support pattern (*Figure 4*). However, I find it still difficult to understand the phase relationship between limbs with regard to swing and stance phase. For example, to what extent does the onset of the swing phase of the forelimb occur during the swing or stance phase of the ipsilateral hindlimb? In order to have a better interpretation of inter-limb coordination, it might be worthwhile to add an additional figure with swing and stance phase indicated for each individual limb*.

Yes, we agree that this is complicated and we had focused on stance phases (Figure 4) together with the support patterns (Figure 4) out of space concerns. There are also changes in stance-swing phasing in *pcd*, which are what cause the changes in support patterns (Figure 7).

Author response image 1.Stance to swing phases for size-matched controls and all *pcd*.**DOI:**
http://dx.doi.org/10.7554/eLife.07892.023

We recognize that this is still complicated. There are more traditional Hildebrand-style plots for *pcd* and size-matched controls, for 3 speed bins (speed increasing from top to bottom panels).

Author response image 2.Average Hildebrand plots aligned to FR stance onset.**DOI:**
http://dx.doi.org/10.7554/eLife.07892.024

We hope this makes it more clear. We have added Hildebrand plots to Figure 4.

Reviewer #3:

*The authors of this study have developed an automated kinematic analysis of locomotion in mice. They use it for locomotor analysis in freely moving wild-type mice. A detailed description of locomotor parameters allowed them to create a mathematical model to predict most of the parameters knowing just the walking speed and the body size of the animal. They extend this analysis to a mouse model for Purkinje cell degeneration (*pcd*). They conclude that the traditional locomotor parameters (stride length, cadence, intra-limb coordination) and forward trajectories are not different in wild-type mice and* pcd *mice*.

This last sentence is not quite accurate and we need to clarify here that we do find that these parameters *are different* in *pcd* mice, but that these *differences disappear when we take changes in walking speed and body size into account*. This is an incredibly important point that changes the entire message of the paper.

*In contrast they conclude that the whole-body coordination is specifically impaired in* pcd *mice. Furthermore, a specific description of nose and tail oscillations in pcd mice were modeled as passive consequences of the forward motion of the hindlimbs. The main conclusions from the study are: a) only by studying whole-body coordination will it be possible to reveal the ataxic locomotor phenotype, and b) the observed locomotor changes are consistent with the hypothesis that the cerebellum provides a forward model for motor control. Although I find the study of great interest there are a number of issues that are problematic for the interpretation of the data. The authors will have to address these issues to avoid making incorrect statements about the findings and the method used*.

Thank you and we address all of these issues, below.

*1) The authors make a big deal out of using machine learning as an 'objective' way of measuring locomotor parameters. But there seems to be a lot of noise in the detection. This is seen in*
Figure 1*. Where is the level set for step onset and offset*?

The swing and stance onsets are calculated from the bottom view, not the side view data and so were plotted in Figure 1. We point out that the noise in the detection was quantified in Figure 1—figure supplement 1 where we compared manual clicking on each paw on each frame with the automated detection for 11 movies. Here you can see a few important points. Importantly, Figure 1, which the reviewer singles out as “noisy” represents vertical (z) axis tracking, which is typically not measured at all in other papers. In the figure supplement you can see that the “z” tracking is noisier than x and y tracking, for several reasons. First, *because of the very small vertical movements of the paws, the movements are only a few pixels high* – the apparent “noise” in the detection in Figure 1 is generally less than 1mm, which corresponds to *2.5 pixels* even on our high resolution movies. Second, because of the relatively large size of the paws compared to the vertical displacement, *most of the apparent discrepancies between manual and automated tracking are within the size of a paw* (all black data points in the supplement). Thus this “noise” is actually largely biological – the paw position is defined as the center of the paw, and as you can see in the movies, the center of the paws moves vertically even during the “stance” phase. Thus it is difficult/moot to say whether the discrepancies between detection and manual clicking in these cases are from automated errors or human ones. Finally, we refer the reviewer to the movies where you can evaluate the automated tracking performance (again, focusing on the bottom view, which forms the basis for all stride analyses, because of these considerations) first hand.

Figure 2
*is an even stronger case. At very low speed (<0.4) the stride length varies with a factor of 10 from 1 cm to 10 cm. 10 cm long stride lengths are not common at these speeds in mice. It must be a detection mistake. Also there are many values with very low stride lengths spread over all speeds. One would like to know how much irrelevant noise that the automated procedure introduces for example by showing traces from the low speed locomotion with 1 cm and 10 cm stride lengths*.

Exactly, these values are not common at these speeds in mice. Please keep in mind that in Figure 2, *nearly 10,000 strides are plotted*. We shared our raw data exactly to emphasize the point of how noisy this unconstrained, virtually unfiltered dataset was, and how important it was to account for this variability. As the reviewer points out, there are some outliers. However, *the assumption that this “noise” must result from tracking mistakes is incorrect*.

The specific examples the reviewer requests, of short and long stride lengths at slow walking speeds, are almost always accounted for by the fact that *we calculated walking speed stride-wise, from stance onset to stance onset*. Because we did not filter our data for continuous walking (because stopping and starting may be an important feature of some motor phenotypes, e.g. basal ganglia), there were sometimes long stance periods when the animal paused just before re-initiating locomotion (Figure 1—figure supplement 1, on the right). Figure 1—figure supplement 1 shows examples (as requested) of very short (left) and very long (right) stride lengths at slow walking speeds, that show that these “outliers” are not, as the reviewer assumes, due to detection mistakes, but rather to sudden changes in walking speed.

On the other hand, we do not claim that our tracking or swing/stance point detection are perfect (see e.g. green points in Figure 1—figure supplement 1). However we point out that the total number of outliers in Figure 2 – including not just tracking failures but also the kinds of biological variability shown in the figures above – is 130 out of 9,602 total strides. Thus *tracking or swing/stance detection errors are affecting at most ∼1% of strides*. Given our multi-level modeling approach to analyzing the data, these points do not pose a significant contamination threat. Further, we think that the major benefits that come from using automation and avoiding data selection and filtering more than compensate for any occasional imperfections.

Everything in Figure 2 is known from many previous studies of rodent locomotion – so there is nothing new about it except that the authors have analyzed a large amount of steps.

This is not true. Please keep in mind *that the thick lines in*
Figure 2
*are the outputs of equations that predict gait parameters as a function of walking speed and body size*, which is entirely new. In the words of Reviewer 2, in Figure 2 “The authors provide an elegant analysis of the correlations between weight, speed and specific parameters of single-limb movement […] The authors conclude that, given body weight and speed, they can make reasonably accurate predictions for single-limb movement parameters.” This is very important – Figure 2 and its supplements show the derivation of equations that predict movement parameters of individual limbs for a mouse of a given size walking at a given speed – and the littermate data in Figure 3 validates these predictions for a new cohort of animals. Without this analysis, which is new, we would not have been able to isolate specific deficits in ataxia.

Moreover, the *3D paw, nose, and tail trajectories during locomotion at a range of speeds in freely walking mice* that we present are, to our knowledge, entirely new. To emphasize this, we have pulled this data out of the supplement to Figure 3 and included the forepaw kinematics in Figure 2 and hindpaw kinematics in Figure 3.

*2) The authors make a point out of not seeing any intralimb differences between wild-type and* pcd *mice. Looking at*
Figure 3
*this referee is not convinced, and it seems that this statement that is likely to be wrong. There seems to be clear differences in almost all parameters. Statistical tests will be needed to support the claims*.

Unfortunately it seems that the reviewer’s misunderstanding of Figure 2 has permeated into this interpretation of Figure 3. To clarify, as we state in multiple places in the text (for example, in the subsection “Changes in gait parameters for individual limbs are predicted by changes in walking speed and body size”), “the gait parameters of individual limbs of *pcd* mice were, overall, quite different” and “although they were different from control mice, individual limb parameters for *pcd* were comparable to the values for control mice of similar body size walking at similar speeds”. Also in the subsection “A quantitative framework for locomotor coordination reveals fundamental features of mouse gait ataxia” we emphasize that “observed differences in traditional gait parameters were a secondary consequence of differences in body size and walking speed”; there are real differences in individual limb movements in *pcd*. However, the thick lines in Figure 3 are model predictions, not fits to the data and so the differences in those graphs the reviewer points out, while real, actually do not reflect ataxia. To make this point more clear we have added a new Figure 3—figure supplement 1 which directly compares *pcd* and size-matched controls and the statistics are provided in panel E (none are significant).

The only panel in Figure 3 where we claim that *pcd* and controls are the same is the forward paw trajectories shown in Figure 3. There are differences between control and *pcd* in Figure 3 – these are *the off-axis movements and joint angles and we fully agree that these reflect ataxia*, which is why we refer specifically to “forward” or “on-axis” trajectories vs. “3D” or “off-axis” trajectories throughout the Results and Discussion. To make this point more clear and separate the results from the forward paw motion results, we have added an additional heading “Single-limb deficits in *pcd* are restricted to off-axis paw trajectories and joint angles” to the relevant section of the Results.

*3) A main conclusion is that the interlimb coordination changes from wild-type to* pcd *mice. More specifically from a trot to a walk pattern. Walking and trotting patterns have been described for tetrapods to be distinct gaits by many different studies. They are visible at different speeds of locomotion. In your experiments wild-type mice exhibited trot as unique gait used at all speeds, even at the lowest ones when the speed is below 0.2 m/s and the support limbs are three or four (*Figure 4*). In contrast the pcd mutant mice exhibit only walk, reaching a maximum speed of 0.25 m/s. What is odd about this difference is that walk is clearly found in wild-type mice by all others than these investigators. I believe that it is present even in the present study but that it has been missed (how can you have trot with 3 or 4 limb on the ground?)*.

We recognize that it is traditional to conceptualize interlimb coordination in terms of gait categories, and we have changed our treatment of this topic in the text to be more clear and to emphasize what we are, and what we are not, trying to claim. However, our perspective differs meaningfully from the reviewer’s on this point. We actually spent quite a long time wrestling with this data and trying to analyze it in terms of categorical gait patterns such as trot and walk (not to mention cruciate vs. alternate patterns a la Cheng et al. 1997 in the rat), but it didn’t work, and we think these plots of stance phase density as a function of walking speed give a sense of why (Figure 4—figure supplement 1).

On the left, the stance phases of all of the strides of size-matched controls are shown in a density plot – values around 0 on the y-axis would be a trot, while values above the x-axis would be a walk. Stride count density is color-coded. The first thing that pops out from this analysis is that *there is no category boundary between a walk and a trot in mice* – the data vary continuously along the x and y axes. We realize that this is different from the case in many tetrapods, and even in flies (Mendes et al, *eLife* 2013) – but this is the data. We agree with the reviewer that mice are more likely to “walk” at lower speeds, but *without a category boundary between walking and trotting, we find that analyzing phase values is the only appropriate quantitative way to deal with this data* – how far from zero would the phase value need to be for us to call it a walk? How close to zero to call it a trot?

We have reworded our language in the paper about wildtype mice to de-emphasize the trot and instead focus on diagonal alternation patterns. We have also included the density plots above (as well as similar L-R phase analysis).

So with this as the background (and we have altered the text and include this figure in the paper), we clarify: we are not trying to overturn existing data or claim categorically that wildtype mice don’t walk – *but what is clear is that* pcd *mice don’t trot*, even when they do locomote at higher speeds.

Further, to answer the question of “how you can have a trot with more than 2 paws on the ground?”, as the reviewer surely knows, and according to the terminology of [27], this depends on the duty cycle and the stance to swing transitions (see Hildebrand plot and stance-to-swing phasing above in response to Reviewer 2). Diagonal pairs can still enter stance simultaneously with either a running trot or a walking trot, and in both cases, the primary support pattern will be 2-paw diagonal. What will change will be whether there is some 0-1 or some 3-4 paw support at the transition points. Our new supplemental figure addresses this directly.

*Since the* pcd *mice locomote slower than the wild-type mice the change in coordination could simply be caused by a change in gait patterns that is speed dependent*.

Actually, no, it can’t. This is very important – *all of the analyses comparing coordination in pcd vs. control were both size- and speed-matched*. And, as we pointed out above, the tail oscillation in *pcd* is successfully modeled as a passive consequence *of the forward motion of the hind limbs only, which does not change in* pcd. Also, we show in the interlimb coordination figure that hind limb double support ratios do not change in *pcd*. We have clarified this in the subsection “Whole-body coordination deficits in *pcd*”.

*It is known for many animal models (mouse as well) that the inter-limb and whole body coordination change when the animal uses different gaits. At the mechanical level different models have been used to explain the different body coordination of walking and trotting gaits (Mechanical work in terrestrial locomotion: two basic mechanisms for minimizing energy expenditure,*
*Cavagna et al., 1977**, American Journal of Physiology). Thus, most of the differences you describe between wild-type and mutant mice could be due just to the comparison of mice using two different gaits that clearly rely on different inter-limb and whole body coordination (e.g.*
Figures 4 and 6*). Indeed you have clear indications that walk is also expressed in wild-type mice at low speed (*Figure 4*). The data should be reanalyzed to address this issue. I am convinced that if the data are reanalysed, walk will show up in wild-type mice as well. This will dramatically change the conclusion of the story*.

The fact that walk does show up in wildtype mice (explained above, and as the reviewer points out you could already see this from our Figure 4) absolutely does not change the conclusions of our paper. All coordination analyses were size- and speed-matched. Moreover, although there are no categorical gait transitions, the coordination results hold across speeds, regardless of whether wild-type gait patterns are more walk-like or more trot-like.

*4) The discussion about the superiority of the present method to others needs to be toned down. The general statement that the automated analysis described in the study provides a robust way for locomotor analysis avoiding "false positive findings that permeate the mouse locomotor literature" is not true and likely to offend the locomotor field at large*.

We sincerely apologize for any offense (which was certainly not intended) and have completely changed our language on this point (see also response to point #4 of Reviewer 2, above). However we also hope that our clarifications about the content of Figures 2 and 3 have helped the reviewer understand what we meant by “false positives” – *the gait parameters are different in* pcd*, but these differences are accounted for by walking speed and body size and therefore do not reflect ataxia per se*.

*I strongly recommend that the authors consider gaits (walk, trot, gallop and bound) in their analysis. Since you have done an incredible work using size-matched controls, it will be helpful if the difference between wild-type and pcd mutant mice account just for the degeneration of the Purkinje cells and not for errors due to a comparison between mice using two different gaits*.

We wrote our response to Major Comment #3, above, before the recent paper by Bellardita and Kiehn (in press), came out. We now also refer directly to the new paper in our Results section (in the subsection “Front-hind Interlimb coordination is specifically impaired in *pcd*”):

“In our experiments, wildtype mice walked in a symmetrical trot pattern across speeds – each diagonal pair of limbs moved together and alternated with the other pair (Figure 4, left). […] This increased instability indicates that *pcd* mice are not simply switching to a more stable gait pattern, but rather, are unable to properly time their front-hind limb movements to generate a stable, efficient gait.”

*5) To my knowledge* pcd *mice showed not only a degeneration of the Purkinje cells but also the slower degeneration of the photoreceptor cells of the retina and mitral cells of the olfactory bulb. Since you have used animals between 41 and 154 days old, could it be that a shorter range of speed is the result of a disorganization of the visual system and not only due to an impairment of the neural control of the locomotion? They walk slower because they are half blind*?

Please see our response (including the new Figure 5—figure supplement 2) to a similar point raised in point #1 of Reviewer 2, above. The new figure shows that results hold in *pcd* mice less than 50 days old, before extracerebellar degeneration. Also, separately, we have measured locomotion in blind mice and we do not see the alterations reported here, and we note that even if they did walk more slowly for this reason, our analyses were all speed matched so this alone could not account for our findings.

[Editors’ note: the author responses to the re-review follow.]

The reviewers have discussed the reviews with one another and the Reviewing editor has drafted this decision to help you prepare a revised submission.

*The reviewers all appreciated the fact that the novel LocoMouse tool permits automated analysis with high resolution of locomotion of freely walking mice, and they judged that this method has the potential to be of value to the field. They provide a number of comments for strengthening the manuscript, which are included below. These comments center on the being more explicit about the limits of the work, its interpretation (especially regarding the mutants), and comparison to other studies; providing more information about certain analyses; and addressing some points for clarity of presentation and communication. Specifically*:

1) Toning down the claims about the superiority and power of the new method, perhaps providing a more explicit comparison to commercial packages;

2) Toning down the claims about providing direct evidence for forward models, perhaps simply by shifting these ideas to the Discussion;

3) Limiting the claims about changes and the lack of changes in the mutant, perhaps by focusing on the idea that one kinematic measure was preserved while others (interlimb and intralimb) were not.

4) Providing a better description of the construction of the passive model, and (if feasible) providing a physical model (or explaining why this was not done);

*5) Addressing the point either experimentally or with clarification of the (apparently) very low N (=3) for the* pcd *mice*;

6) Improving presentation of the results/figures/sequence of calling figures, as indicated in the reviews.

We sincerely thank the reviewers and editors for their thoughtful and constructive comments on our manuscript. We address each of these six points and our corresponding revisions in detail below, in response to the specific concerns raised by the reviewers.

Reviewer #1:

*1) The authors emphasize in several places that 'traditional' measures of kinematics are unimpaired in mutants. I found this to be overly-simplistic, at least based on the results presented here. As I understand it, the main consistency is in the x-axis forward paw motion, while there are changes in vertical and mediolateral paw motion and in joint angles. Changes in vertical paw trajectory and joint angles are hardly 'non-traditional', 'complex', or even '3D' and all measures are definitely 'individual-limb'. This point is made several times in the text (see points below) and seems unwarranted. I'm not saying the results aren't interesting, nor do I think that these conclusions are necessary for the impact of the paper – I just don't think that this conclusion as stated is supported*.

We apologize for not being more clear. We do not intend to claim that individual limb movements are intact in *pcd* mutants (they clearly are not). The reviewer is correct that we were trying to emphasize the preservation of forward motion of the paws. Our use of the subjective term “traditional” was misleading and we have removed it. We were using “traditional” to refer to stance-based analyses from paw print analyses and popular commercial systems for mouse locomotion such as CatWalk. However, clearly joint measurements and vertical trajectories (both affected in *pcd*) are traditional measures for ataxia in other species. We have clarified our terminology throughout the text to reflect more accurately our intended meaning, which is that the parameters that are usually reported as evidence for gait ataxia in mice were only different in *pcd* in ways that had nothing to do with ataxia (because they were associated with changes in body size and walking speed). Specifically, we now refer to the parameters depicted in Figure 2 as “basic stride parameters” rather than “traditional gait parameters”, because not only was the use of “traditional” problematic, but there is a critical distinction between the effects of ataxia on strides (Figures 2 and 3) and gait (which also includes the interlimb and whole-body coordination shown in Figures 4, 5 and 6, for example). When we refer to both the stride parameters from Figure 2 and the forward trajectories of the paws shown in Figure 3 (such as in the Abstract) we have replaced “traditional, individual-limb parameters” with “forward motion of individual paws.”

*2) On a similar point, it's not clear to me that any of their measures are really looking at within-limb coordination, at least as this is usually considered. Most of the measures relate to paw movement, whereas traditionally issues of coordination (e.g. in studies examining the role of the cerebellum, as cited in the paper) consider interactions between joints. Yet these analyses really aren't done here. They show that joint angles are apparently different in the animals, but these are for single joints as opposed to joint angle relationships (and what those angles are is unclear, see below). I don't think that the results shown here rule out a possible deficit in traditional, within-limb coordination. I think this is mainly an issue of presentation, however, rather than one of substance. As stated above, I don't think the impact of the paper relies on this conclusion and it could be made more precise without loss of significance*.

Indeed, this was also an issue of clarity on our part – we do not intend to rule out intralimb coordination deficits in *pcd*. In fact we think the altered y and z trajectories and restricted joint angles do represent deficits in within-limb coordination. The changes in language described in response to point 1, above, should help clarify this.

*3) I potentially like the analysis of nose and tail passive movements very much, but I did have a few issues with it. First, although the results are consistent with a role of the cerebellum in predictive control based on forward models, they're not really a direct demonstration thereof. A more direct demonstration might involve examining adaptations to perturbations, e.g. adding a weight to the tail and seeing that initially control animals look like mutants but then learn to predict and compensate for this, whereas mutants never do so. Also, could the results simply be interpreted as one of a limited capacity for motor control? e.g. mutants have limited capacity for producing locomotion and expend most of it on x-axis movement of the paw since it's critical for task performance, whereas other aspects of behavior are allowed to vary? I appreciate the authors are careful on this point to say that the results are 'consistent' with this idea, but alternative explanations might be considered more directly*.

We have toned down and revised the forward model interpretation in response to this and other comments from the reviewers. To address the reviewer’s point directly, we don’t think the alternative of a limited capacity for motor control in *pcd* can fully explain our data. First, there is generally not an increase in variability in *pcd*. Second, although it could explain why the tail/nose movements are larger in *pcd*, we don’t see how it could explain the systematic shift from phase locking/leading in control animals to phase lags (that are consistent with time-delayed movements) in *pcd,* a phenomenon that we observed not just in the nose/tail movements but also in the patterns of interlimb coordination.

*4) For the nose/tail passive analysis, I originally thought that this was an actual physical model, rather than the geometric model used here. It seems clear that a physical model is the better way to do this. Why wasn't it done? The geometric model, with its assumptions of lags taken from data which is actually observed, seems more indirect and therefore less compelling*.

Our intuition was initially the same as the reviewer’s and we actually began by constructing a physical model. However, the physical modeling required us to make multiple assumptions about the biomechanics of the tail, and we wanted to avoid simply tweaking parameters to achieve a model that would fit the data. We therefore decided to ask whether a very simple geometric model with as few assumptions as possible might do just as well. The passive geometrical model had just two assumptions. First, that passive nose and tail movements would result directly from orthogonal coupling to the forward oscillations of the hind paws – this was the key insight. Second, that these movements would occur time-delayed relative to the hind paw oscillation (and that the time delays would be longest for the mouse’s rump and become progressively smaller for each more distal segment, due to decreasing mass). It is readily apparent from the data (Figure 5, Figure 5—figure supplement 1) that the assumption of time-delayed movements only holds true for *pcd* mice, and not controls. However the correspondence between the passive model predictions and the nose and tail movements of *pcd* mice was also striking at the level of the waveforms themselves (Figure 5). We were surprised at how well this simple model fit the *pcd* data and found the simplicity of this approach to be quite powerful. However given that biomechanics can be counterintuitive we have also implemented a simplified, 3-segment physical model of the tail as a proof of principle. In Figure 9 we show the results of this physical model, which are in general agreement with those of the geometrical model.

Author response image 3.Physical model.The physical model of the tail consists of three masses 1, 2 and 3 (the later being the most distal) connected via an angualar spring-damper system. The masses of each segment were estimated from measurements of a real mouse tail (10g, 0.6g, 0.001g for segments 1, 2 and 3 respectively). The springs and dampers were estimated manually (k_1_=13e-6 Nm/deg, c_1_=0.95e-6 Nms/deg, k_2_=5e-6 Nm/deg, c_2_=0.7e-6 Nms/deg and k_3_=20e-6 Nm/deg, c_3_=0.2e-6 Nms/deg). The model was implemented and tested in SimMechanics MATLAB 2015a. **(A)** The phases of segments 1, 2 and 3 in the model (lines) superimposed over segments of 18 and 15 of the *pcd* tail. **(B)** Trajectories of tail segment 8 for controls (black), and *pcd* (purple), and tail segment 2 for the physical model.**DOI:**
http://dx.doi.org/10.7554/eLife.07892.025

*Finally, as I understand it (and I might have missed it), this analysis was only done for the mutants. It would be very good to have done the same analysis for the control animals and show that the tail/nose movements are inconsistent with passive movement. E.g. it could have been that the movement of the pelvic girdle was different in controls, leading to less tail movement (even though the hindlimbs are in alternation for both controls and mutants)*.

This is an important point and we were not as clear about it as we should have been. We have revised the Materials and methods section and now provide more detail about the geometric model. The stride parameters were taken from the statistical stride models in Figure 2, for a small mouse (15g). Because these stride parameters were indistinguishable between *pcd* and small controls, the key point here is that the passive geometric model is not a model for either the mutants or the controls. The model traces in Figure 5 are based on stride data that did not vary by genotype, plus the fixed time delays. The model fits the *pcd* data better than control data because only the *pcd* tail/nose moved time-delayed and directly orthogonally-coupled to the hind paws. Also note that in terms of differences in pelvic girdle between the two mice, the only factor that could influence the amplitude of tail movement in our model would be the wider base of support in *pcd* (Figure 3) – which would actually predict that *pcd* mice would exhibit smaller amplitude tail oscillations than controls (because of smaller angular excursion of the hind paw axis). Further, this would not account for any differences in phasing between the two groups of mice.

Reviewer #2:

*The paper by Machado et al. examines gait patterns in wildtype and Purkinje cell degeneration (*pcd*) mice using a novel system for tracking locomotor kinematics in freely walking mice. The authors conclude that the individual limb kinematics of* pcd *mice are normal when the weight and the walking speed of the mice are taken into account. In contrast,* pcd *mice have impaired coordination of movements across joints, limbs and body. These results are interpreted as evidence that the cerebellum provides a forward model. This is a strong paper. I commend the authors for wrestling with a complex dataset and extracting from it a number of thought-provoking findings. I personally found many of the results exciting, but I must admit that I'm not an expert in the field of locomotion or ataxia. It was not always clear if a particular result in the paper was a major advance or just confirmation of something we already knew. For this reason, I think that it's important for the authors to clarify and emphasize the significance of their results throughout the text. I've given two examples below*:

Thank you. We have rewritten several sections to be more explicit about the specific findings of previous studies and to provide context for our results (see specific instances below).

*1) It is remarkable that in wildtype mice, one can account for upwards of 80% of the variance in kinematic parameters like stride length, swing velocity and cadence, simply by taking into account the walking speed and the weight of the mouse. Is this a new approach? In their rebuttal, the authors mention that the*
[11]
*paper found that all the gait differences between lurcher and control mice disappeared when they corrected for walking speed. Did Cendelin et al. make the correction using a similar modeling approach*?

No, they did not. [11] used the CatWalk system to measure stride parameters in visibly ataxic lurcher mice and found while they were different in lurchers and controls overall, the values for the fastest lurcher trials and the slowest control trials were comparable. They did not deal with body weight and they used speed grouping rather than statistical modeling to identify speed effects. They hypothesized that CatWalk analysis was not able to detect gait ataxia because it resulted from “titubations, pro-, retro- and latero-pulsions” not captured by CatWalk, while “the coordination of limb movements is affected minimally” in ataxia. Our finding that the differences in stride parameters in *pcd* can be accounted for by changes in walking speed and body size is consistent with their results. We extend this finding by providing quantitative predictions for these stride parameters across mice and trials, incorporating body weight, and analyzing swing kinematics, interlimb coordination, and head and tail movements to identify which of these features truly represent ataxia vs. changes in walking speed and body size. In the end we reach conclusions that differ from their predictions regarding the nature of ataxia, but their study and their statement, “The fact that almost all differences in gait parameters are markedly dependent on walking speed seems surprising because the difference between a walking Lurcher mutant and a wild type mouse is clearly visible with the naked eye”, provided a major motivation for our study – to see if we could quantitatively capture the fundamental features of ataxia in freely walking mice.

[6] used a mixed-effects modeling approach to analyze the relationship between stride parameters and walking speed, but did not assess the other possible factors of body size, gender, age, etc., nor did they consider ataxia. Also to our knowledge no one has presented predictive models for stride parameters as we do here and we think this, together with the large dataset we provide, is a useful resource for future mouse locomotion studies.

We have revised several sections of the manuscript (notably the rewritten Introduction) to put our findings more explicitly within the context of these two important previous studies.

*2) One of the main findings is that individual limb movements of* pcd *mice are relatively spared (when weight and walking speed are taken into account), whereas interlimb coordination is impaired. It is not clear how much of this we already knew. For example, in the subsection “Front-hind Interlimb coordination is specifically impaired in* pcd*”, the authors mention that previous studies in ataxic mice have failed to detect any impairments in the kinematic parameters of individual limbs. Did those studies not examine interlimb coordination as well? If the authors are the first to show that mice with cerebellar deficits have a specific impairment in locomotor coordination but not in movement of individual limbs, this is a big deal*.

Nearly all papers using overground locomotion to assess ataxia in mice have reported differences in paw stride parameters (which are real, but are artifacts of walking speed and body size) at face value. The reference was to the [11] paper just described, which is unusual in its consideration of walking speed. Other papers have shown deficits in both stride parameters of individual limbs and interlimb coordination in ataxic mutants (e.g. Vinueza-Veloz 2014). To our knowledge we are the first to isolate impairments in coordination (across joints, limbs, and body) from the non-specific changes in stride parameters and forward motion of the paws.

Additional recommendations to help make an already strong paper even stronger:

*3) Drop the emphasis on forward models. The authors argue that the deficits they've uncovered point to a lack of a forward model in pcd mice. Although I found some of the evidence presented intriguing (for example the analysis of tail movements), it is far from conclusive. Furthermore, it is difficult to reconcile the normal individual limb kinematics with a faulty forward model – individual limbs are controlled by multiple agonist/antagonist muscles whose coordination requires well-calibrated forward models. I think the authors can still interpret their results in the context of forward/inverse models, but I strongly recommend that they save this interpretation for the Discussion, tone it down a bit and address the limitations. The paper would be stronger if the second paragraph of the Introduction is rewritten to highlight the main questions to be addressed in the paper, instead of offering to resolve controversies about forward vs inverse cerebellar models*.

We have moved the treatment of internal models to the Discussion, edited it (both toning down and shortening), and rewritten the second paragraph of the Introduction as requested.

*4) Analysis of variability: All the analyses are based on average data. For example,*
Figure 3
*shows that there is no difference in the individual limb kinematics of* pcd *mice and sized-matched controls. But this is done only for the average kinematic values. Is it not possible that* pcd *mice might show increased trial-to-trial variability in their movements relative to sized-matched controls? How would such a finding change the authors' conclusions*?

We, like the reviewer, fully expected to observe increased variability in *pcd*. However, aside from the increased variability in hind paw placement that we reported in Figure 4, we actually found that the movements of *pcd* mice were either equally, or in several cases, even less variable than controls. To be honest we didn’t know what to make of this finding, and didn’t want to make strong claims about a surprising lack of variability. However leaving it out was probably not the best choice either, and so we have now included the other variability analyses (including those corresponding to Figure 3) in the text of the paper. (Also note that the density plots, now in Figure 4—figure supplement 1, do show the full range of data for limb phasing.)

Reviewer #3:

*The authors developed a machine learning algorithm to track the individual limbs as well as the nose and tail of mice in 3D during natural, voluntary locomotion. They provide a very detailed analysis of locomotion parameters in control and ataxic Purkinje cell degeneration (*pcd*) mice, revealing that individual limb kinematics in* pcd *mice are not different from control mice when accounting for the mouse's weight and speed, but that* pcd *mice present multi-joint and inter-limb-coordination deficits. The study has been carefully conducted and provides new and useful tools for measuring and analyzing specific features of locomotion and ataxia with unprecedented accuracy*.

Thank you.

*1) The construction of the "passive model" (*Figure 5*) is unclear. For example, does this model take into account the mass of the tail? Please describe in more detail how the passive prediction is calculated*.

We rewrote the Materials and methods paragraph describing the model. Please also see the responses to Reviewer 1 point 4, above.

*2) The authors contrast the forward model with an inverse kinematic model in the Introduction. What do they think the inverse kinematic model would predict and how does this contrast with their findings*?

We expected that a failure of an inverse kinematic model would cause changes in forward paw trajectories, which were surprisingly intact in *pcd* (Figure 3). We have now de-emphasized the internal model interpretation in part by moving it to the Discussion as suggested above.

[Editors' note: further revisions were requested prior to acceptance, as described below.]

There are only a few remaining issues that need to be addressed before acceptance, as outlined below:

*In particular, the manual joint-angle analysis shown in*
Figure 3
*seems limited by the single animal in each condition, and discussion among reviewers and editors led to the consensus that it could simply be dropped without major impact on the manuscript. We therefore recommend cutting this one measurement and illustration. Alternatively, if there is some reason why a single animal in each category is of value, please justify the experiment clearly. This point is explained further below*.

We have removed the joint angle analysis from the paper.

*In the subsection “Individual limb parameters”: This description makes it seem like one* pcd *animal and one control were used for this analysis. If so, this seems very questionable and underpowered. I'd be willing to bet that, with enough n, it would be possible to find a significant difference between two different weight matched controls. I'm not sure how the stats behave in this situation, but it's not clear to me how mixed models would be able to partition variance due to random effects (individual subjects) from the fixed effects (mutation) if there were only one animal in each group. It seems that for other comparisons with weight matched controls much larger N's are used, whereas this analysis is limited because of the need to hand mark frames. It might be easiest to simply drop this joint angle analysis – I don't think its omission would change the impact of the paper. Also, shouldn't it be 'knee, heel, toe’? And here it's stated that the angle of the foot and arm relative to the ground are quantified – I might have missed it, but I wasn't sure where that was reported in the Results*.

We have removed the joint angle analysis from the paper.